# Towards verifiable text generation with symbolic references

**Lucas Torroba Hennigen**◇,*   **Shannon Zejiang Shen**◇,*   **Aniruddha Nrusimha**◇
**Bernhard Gapp**†   **David Sontag**◇   **Yoon Kim**◇
◇Massachusetts Institute of Technology   †Good Data Initiative
lucastor@mit.edu zjshen@mit.edu anin@mit.edu
bernhard.gapp@gooddatainitiative.com dsontag@mit.edu yoonkim@mit.edu

## Abstract

LLMs are vulnerable to hallucinations, and thus their outputs generally require laborious human verification for high-stakes applications. To this end, we propose *symbolically grounded generation* (SymGen) as a simple approach for enabling easier manual validation of an LLM's output. SymGen prompts an LLM to interleave its regular output text with explicit symbolic references to fields present in some conditioning data (e.g., a table in JSON format). The references can be used to display the provenance of different spans of text in the generation, reducing the effort required for manual verification. Across a range of data-to-text and question-answering experiments, we find that LLMs are able to directly output text that makes use of accurate symbolic references while maintaining fluency and factuality. In a human study we further find that such annotations can streamline human verification of machine-generated text.

⊙ https://symgen.github.io

## 1 Introduction

Many applications of generative AI involve generating text based on structured data (e.g, tabular data, surveys, API outputs, etc.) that is known (or assumed) to be trustworthy. For example, newspapers may be interested in generating summaries of sports games based on official match statistics, and search providers would want generative search engines to ground its output in search results. These applications require *conditional text generation* that is fluent, accurate, and verifiable with respect to the conditioning information. Large language models (LLMs) have advanced to the point where they can sometimes be relied upon to generate fluent- and faithful-enough summaries of text and other structured data. However they remain far from perfect (Liu et al., 2023; Yue et al., 2023), and insofar as high-stakes applications will almost certainly require a human to manually *verify* that an LLM's generation is accurate with respect to its conditioning information, there is a need to develop frameworks that maintain the fluent and accurate few-shot generation capabilities of LLMs while enabling streamlined human verification of the model's output.

This paper proposes *symbolically grounded generation* (SymGen) as a step towards verifiable text generation with LLMs. Given a string representation of data in a structured but human-readable format (e.g., JSON, YAML, etc.), we prompt the LLM to generate the output with explicit symbolic references into the provided data structure instead of generating the text directly. The references are then rendered using a parser, which can faithfully represent values from the original data, and can moreover be used to include visual cues that enable a user to readily verify the provenance of a particular span of text (see Fig. 2 for an overview). In contrast to classic templated approaches for text generation (Kukich, 1983; McKeown, 1992; McRoy et al., 2000), SymGen instead offloads the template specification process to an LLM. We exploit the fact that LLMs have likely been sufficiently exposed to such kind of templated text during pretraining that they are able perform zero- and few-shot text generation with symbolic references across multiple domains.

---

* Equal contribution.

To demonstrate its effectiveness, we evaluate SymGen on its ability to retain the same textual quality as regular generation while improving the ability of users to verify the final LLM response. Specifically, we study SymGen's performance along four axes and across five datasets: *textual quality* (§3), *citation accuracy* (§4.1), *helpfulness of SymGen citations* (§4.2), and effectiveness in downstream extensions of SymGen to question–answering and mathematical reasoning (§5). Overall, our experiments on zero- and few-shot data-to-text generation and question answering indicate that LLMs can be prompted to generate fluent text that is interleaved with symbolic references which explicitly refer to data fields. Further, we find that the annotations enabled by SymGen simplify the process of post-hoc verification, both as perceived by the user and by reducing the average verification time by 20%.

## 2 Text generation with symbolic references

**Task formulation.** Let $\mathcal{V}$ be some vocabulary, and let $\mathcal{F}$ be a set of fields in a data structure. One special case of conditional generation consists of taking as input some structured data encoded as field–value tuples $d = \{(f_i, v_i)\}_{i=1}^{n}$ with field $f_i \in \mathcal{F}$, value $v_i \in \mathcal{V}^*$, and some instruction $x \in \mathcal{V}^*$, and then producing an appropriate response $y = [w_1, \ldots, w_n] \in \mathcal{V}^*$. One simple technique for this task is to encode the field–value tuples as regular text (e.g., as a JSON dictionary as in Fig. 2 (1)) and provide it to an LLM alongside an instruction (e.g., "Summarize a basketball match given its statistics in JSON.").

**Symbolically grounded generation.** In this paper, we propose a simple *symbolically grounded generation* (SymGen) approach for such tasks. The idea is to first generate a response in *symbolic form* $\tilde{y} \in (\mathcal{V} \cup \mathcal{F})^*$, which interleaves regular text and references to fields from the data. For example, in Fig. 2, the response in (2) would be:

$$\tilde{y} = [\text{The}, \texttt{visitor.city}, \ldots, ).].$$

We can subsequently *render* $\tilde{y}$ by substituting each field $f_j \in \mathcal{F}$ with its corresponding value $v_j \in \mathcal{V}^*$ in the data to obtain the final response. For example, in Fig. 2 the symbolic form (2) is rendered into $y =$

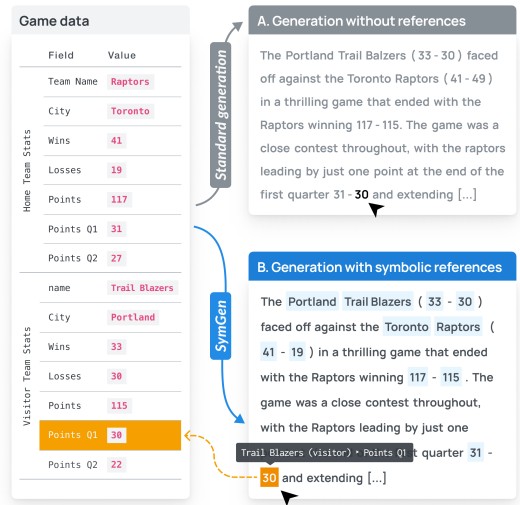

Figure 1: Compare a standard LLM-generated (A) with a SymGen (B, ours) description of a basketball game, based on match statistics. SymGen imbues spans of generated text (highlighted in blue) with symbolic references to the source data, enabling easier verification: e.g., when hovering over a span, the number "30" displays a tooltip and link (highlighted in yellow) indicating the value it is referencing.

[The, Portland, ..., ).], which is shown in (3). Importantly, the inclusion of symbolic references enables fine-grained attribution of portions of the text to their generating fields in the data. In this sense, SymGen is similar in spirit to citation-enabled LLMs (Liu et al., 2023; Yue et al., 2023; Gao et al., 2023), except that our approach focuses on providing very precise and easy-to-validate citations. We stress that our aim with SymGen is not to improve the overall quality of the rendered response, but instead to ensure its textual quality is maintained (§3) while improving the ability of end-users to verify its correctness (§4).

**Operationalization.** While there are many ways of operationalizing the approach above, in practice we have found that prompted LLMs are highly effective at this task. In general, our strategy consists of prompting the LLM to generate its responses in a hybrid form, where fields in the JSON data (which is itself included at the beginning of the prompt) should be referenced using a Jinja-like syntax, i.e., the LLM should generate `{{ field }}`

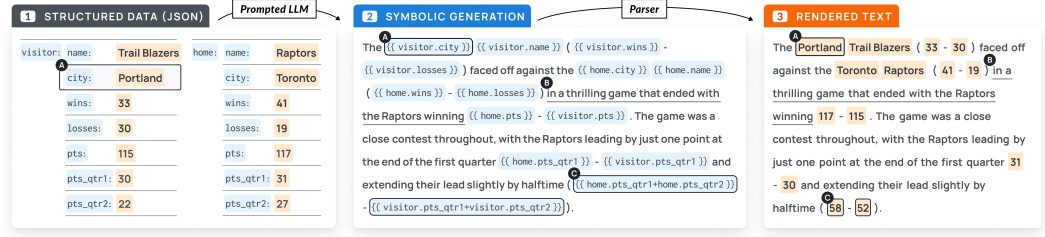

Figure 2: Overview of the proposed symbolically grounded generation (SymGen) pipeline, on an example generated by GPT-4. Given the structured data input (1), we use a prompted LLM to generate a response (2) that contains symbolic references into the data (e.g., the variable `visitor.city` marked by Ⓐ). A parser then substitutes the symbolic references with their corresponding values in the structured data (e.g., `Portland` for `visitor.city`) to obtain the final rendered text (3). Our SymGen pipeline can implicitly leverage the input data to generate prose without any symbolic references (e.g., the span marked by Ⓑ). It can also transform existing variables (e.g., adding up two numbers, marked by Ⓒ) to create unavailable data in the source, while preserving references to the original variables.

to refer to a specific field in the data as opposed to generating the field's value as it would do by default. We can then use an off-the-shelf Jinja parser[1] to render the final text, with minor postprocessing (e.g., based on regular expressions) to make sure we preserve an alignment between rendered references and the field they refer to. Specifically, we consider two methods based of the above:

1. **Direct SymGen**: As described above, we encode the structured data as a JSON dictionary, and then prompt the LLM to generate its output using Jinja-like syntax.

2. **Indirect SymGen**: We also explore a variant of the above, where we prompt an LLM to first generate the response directly *without* any symbolic references, and afterwards prompt it to convert this response into one with a Jinja-like syntax. Effectively, this consists of first tasking the LLM with building a plan, and then conditioning on this plan to obtain the symbolically grounded response.

We detail the used prompts in App. I.2. The advantage of the indirect approach is that it should maintain the same text quality as regular (non-symbolic) generation as long as converting regular prose into symbolically-augmented text is no harder than generating the regular text in the first place; the caveat is that this approach can be more costly with APIs that charge on a per-token basis, as it requires roughly twice the amount of tokens to be generated. We also found this strategy to be unreliable with weaker LLMs such as GPT-3.5,[2] so we only employ this with GPT-4.

This is one of many possible of operationalizations of this method. In particular, we also explored alternatives to Jinja-like references (e.g., Python f-strings for the reference format), and alternatives to JSON (e.g., YAML). We found that the LLMs we tested were able to also work with these formats, which suggests that further gains may be possible by exploring other practical operationalizations SymGen.

**Evaluation.** As mentioned earlier, the main aim of SymGen is to enhance verifiability of the final text without compromising the textual quality over the final generation. To determine whether this is the case, we study SymGen's performance along four axes. The first is whether the *textual quality* of the final, rendered text is comparable to the quality of the text generated by a standard, prompted LLM (§3). Then we evaluate the *citation accuracy*,

---

[1] For example, https://jinja.palletsprojects.com/en/3.1.x/extensions/. Using a parser also allows us to gracefully handle syntax errors made by the LLM, e.g., in the event that a Jinja expression that cannot be rendered, one can resolve it to a default value, e.g., "undefined."

[2] Specifically, we found that GPT-3.5 would often behave as in the direct SymGen strategy, despite being prompted and given examples of the desired, indirect SymGen behavior; see §3.1.

| | | | BLEU | BERT F1 | ER | GER |
|---|---|---|---|---|---|---|
| GPT-3.5-4K | 0-shot | Baseline | 31.15 | 53.43 | 0.00 | 0.00 |
| | | Direct | 26.64 | 47.88 | 0.28 | 0.00 |
| | 2-shot | Baseline | 30.98 | 54.66 | 0.00 | 0.00 |
| | | Direct | 32.43 | 53.32 | 3.30 | 0.09 |
| GPT-4-8K | 0-shot | Baseline | 30.08 | 54.27 | 0.00 | 0.00 |
| | | Direct | 33.31 | 54.07 | 0.00 | 0.00 |
| | | Indirect | 35.02 | 55.58 | 0.00 | 0.00 |
| | 2-shot | Baseline | 32.43 | 56.39 | 0.00 | 0.00 |
| | | Direct | 36.83 | 57.71 | 2.11 | 0.00 |
| | | Indirect | 37.56 | 57.88 | 3.21 | 0.00 |

(a) Results on SynthBio (Yuan et al., 2021).

| | | | BLEU | ROU.-L | ER | GER |
|---|---|---|---|---|---|---|
| GPT-3.5-16K | 0-shot | Baseline | 5.87 | 19.52 | 0.00 | 0.00 |
| | | Direct | 5.07 | 19.14 | 4.00 | 0.00 |
| | 1-shot | Baseline | 12.26 | 22.66 | 0.00 | 0.00 |
| | | Direct | 9.10 | 20.65 | 0.00 | 0.00 |
| GPT-4-32K | 0-shot | Baseline | 4.94 | 19.91 | 0.00 | 0.00 |
| | | Direct | 2.24 | 17.88 | 0.00 | 0.00 |
| | | Indirect | 4.66 | 19.61 | 0.00 | 0.00 |
| | 1-shot | Baseline | 9.11 | 22.16 | 0.00 | 0.00 |
| | | Direct | 7.02 | 21.78 | 0.00 | 0.00 |
| | | Indirect | 8.29 | 21.33 | 2.00 | 2.00 |

(b) Results on Rotowire (Wiseman et al., 2017).

Table 1: For SynthBio, we report BLEU (Papineni et al., 2002) and BERTScore F1 (BERT F1; Zhang et al., 2020) against the reference biographies. For Rotowire, we report the BLEU and ROUGE (ROU.-L; Lin, 2004) of the generated answers against the reference generations in the dataset. We also report the (general) error rate (ER) and the global error rate (GER).

i.e., whether the symbolic references introduced by SymGen are not only correct but also non-trivial in the sense that they cannot be added post-hoc by a naïve baseline (§4.1), and we further assess whether these links actually enable better verifiability in terms of speed and accuracy (§4.2) via human evaluation. Finally, we explore some extensions of SymGen to question–answering and mathematical reasoning (§5). Collectively across all studies, our analysis is conducted on five datasets, which are shown in Tab. 8 of App. I.1. Unless otherwise stated, in all cases we compare against a baseline that consists of prompting the LLM to generate the response directly, without any symbolic references.

## 3 Textual quality study

The first empirical study in this work aims to understand whether SymGen generations exhibit the same quality as a (prompted) LLM baseline; our main goal is not to improve upon this baseline (although in some cases we observe improvements) as our overarching goal is to improve generation verifiability without sacrificing performance. For this set of experiments, we evaluate on two traditional data-to-text datasets (§3.1), and on a new counterfactual obituary generation (§3.2) dataset. For the former, our focus is in analyzing the fluency of the resulting text, whereas in the latter we focus on the factuality of the generated text. We consider both zero- and few-shot settings for each of those tasks.

Depending on the nature of the task (e.g., complexity, size of the structured data, etc.), we evaluate our approach on a mix of GPT-3.5 (with 4K or 16K context windows) and GPT-4 (with 8K or 32K context windows). The full experimental setups, including the prompts, are given in App. I. Anecdotally, we found that extensive experiments with different prompts were not necessary to get LLMs to perform symbolic generation; we expect that further improvements are possible with more prompt engineering.

### 3.1 Data-to-text

SymGen is similar in spirit to data-to-text methods, where the goal is to generate some text based on structured data. Classical approaches to this problem consist of designing a template based on the schema of the data, and populating it with the values of a datapoint at runtime (Kukich, 1983; McKeown, 1992; McRoy et al., 2000). However, relying solely on the schema of the data tends to yield formulaic text. Neural approaches to data-to-text generation (Wiseman et al., 2017; 2018; Wang, 2019; Yin & Wan, 2022) improve on this by generating text that is datapoint-dependent, but achieve this by training on data-to-text

|         |          | EM (%) | ROU.-1 | ROU.-2 | ROU.-L | UR    | ER   | GER  |
|---------|----------|--------|--------|--------|--------|-------|------|------|
| 0-shot  | Baseline | 69.05  | 75.25  | 53.76  | 75.23  | 14.10 | 0.00 | 0.00 |
|         | Direct   | 71.83  | 76.51  | 53.89  | 76.50  | 14.60 | 0.00 | 0.00 |
| 2-shot  | Baseline | 79.34  | 83.97  | 59.87  | 83.97  | 5.53  | 0.00 | 0.00 |
|         | Direct   | 78.61  | 82.90  | 58.15  | 82.88  | 6.76  | 1.39 | 0.00 |

Table 2: GPT-3.5-4K results of the automated question–answering evaluation the obituary dataset, which includes counterfactually generated entities. We report the exact match accuracy (EM (%)) and ROUGE (ROU.-1, ROU.-2, ROU.-L; Lin, 2004) of the inferred answers from the generated text against the true answer given by the source JSON. We also report the percentage of answers that the QA model did not find an answer for (the unknown rate; UR), and the regular (ER) and global (GER) error rates. Refer to §3.2 for a discussion.

datasets. To this end, we revisit this setting in the context of zero- and few-shot data-to-text generation with LLMs.

**Datasets.**  We first consider SynthBio (Yuan et al., 2021), a collection of synthetically constructed *fictional* entities, described in terms of a collection of key–value pairs, which has been used in prior work on templated generation with finetuned models (Zhang et al., 2022). The task is to generate a textual description of the entity. We also consider the Rotowire (Wiseman et al., 2017) dataset, where the objective is to generate a summary of a basketball game given its box (individual player statistics) and line (aggregate team statistics) scores. The Rotowire dataset presents a challenging testbed for our approach, since its data structure is substantially more complex than the previous task (i.e., there are many more fields in each datapoint) and it relies on more specific in-domain knowledge (i.e., understanding how to read basketball box scores charts). To keep LLM API costs manageable, we sample 100 examples from the test set to evaluate on. Refer to App. I for more details.

**Experimental details.**  On SynthBio, we generate a short biography for each entity in the test set using GPT-3.5-4K and GPT-4-8K, and we evaluate against reference texts using BLEU (Papineni et al., 2002) and BERTScore F1 (Zhang et al., 2020). We consider both zero- and 2-shot learning. See App. I.2.1 for the prompts and example generations. On Rotowire, we explore both GPT-3.5-16K and GPT-4-32K (the longer context windows are needed due to the length of the JSON), and evaluate against reference texts using using BLEU and ROUGE (Lin, 2004). See App. I.2.2 for the Rotowire prompts and examples. We experimented with direct and in-direct strategies on both GPT-4-8K and GPT-3.5-4K models. We found that for many examples GPT-3.5-4K would ignore the instructions/in-context examples and directly output a response with symbolic references in the first step. Though the metrics were still reasonable (e.g., for SynthBio, BLEU was 32.67 for 0-shot and 33.10 for 2-shot), we omit these results since they do not reflect the correct strategy.

**Results.**  Our results are shown in the respective tables for each dataset (SynthBio, Tab. 1a; Rotowire, Tab. 1b). The results for data-to-text generation are generally positive. For SynthBio, in the few-shot case, we find that both symbolic generation strategies we considered yield comparable or superior performance to the baseline, indicating that SymGen indeed does not degrade the textual quality of the final generation.[3] Interestingly, in the zero-shot GPT-3.5-4K case, direct symbolic generation underperforms the baseline, though this gap is bridged via in-context learning. This could suggest that biographical generation in symbolic space is slightly harder than regular (non-symbolic) generation for GPT-3.5-4K.[4]

---

[3]For comparison, TempLM (Zhang et al., 2022)—which finetunes a pretrained LLM on the full training set—attains 40.30 BLEU and 54.30 in BERTScore, which are comparable to our GPT-4 results.

[4]For SynthBio, JSON values are not ideally formatted, meaning that LLMs will either output grammatically incorrect text or have to perform post-processing on-the-fly via more complex templates, which seems to increase the likelihood of errors. Refer to App. I.2.1 for an example of how more complex Jinja templates can mitigate this.

For Rotowire, we find that the quality of symbolic generation generally trails the baseline, though this difference is smaller when using GPT-4-32K and indirect generation. Interestingly, GPT-3.5-16K seems to outperform GPT-4-32K in this task, as evidenced by comparing the baseline results under each model, which suggests that better performance may be obtained by modifying the indirect strategy so that GPT-3.5-16K is used to generate summaries in regular prose and GPT-4-32K is used to rewrite them using symbolic references.

In many cases, we find that poor symbolic generations arise from rendering errors. To this end, we also report the percentage of outputs whose symbolic form has at least one error (a specific reference in a response failed to render locally and was instead rendered as "undefined"; ER) and with at least one global error (where the Jinja parser simply failed to run, causing the whole response to fail and be replaced with "The text is not available."; GER). We find that (i) GPT-3.5 tends to commit more errors than GPT-4 and (ii) providing few-shot examples tends to reduce errors, both of which are expected, though we also find that the indirect SymGen strategy leads to more errors than the direct strategy. We believe this to be because adapting regular, non-symbolic text to have symbolic references can sometimes be hard if the JSON is incomplete or the original text is not written in a way that is amenable to the insertion of references.

## 3.2 Counterfactual text generation

There is evidence that LLMs are capable of memorizing their training data (Carlini et al., 2019; 2023) and that this leads to their struggling to generate *counterfactual* data that goes against their learnt priors (Hernandez et al., 2023). In this section, we explore whether SymGen retains the same performance as the baseline when some of the conditioning data is counterfactual in nature.

**Dataset.** We collect a dataset comprised of 167 famous scientists who lived between 1800 CE and 2000 CE (e.g., Albert Einstein, Carl Sagan, Louis Pasteur, etc.) and further generate *counterfactual* variants of each entity. The counterfactual variant is designed to test for the extent to which an LLM may ignore data that contradicts the information an LLM has seen during training, which has been noted to be a common failure mode of smaller language models (Hernandez et al., 2023). For more details refer to App. I.

**Experimental setup.** We use GPT-3.5-4K to generate obituaries for each of these (possibly counterfactual) entities. For the few-shot experiments, we provide two examples (see App. I.2.3 for our prompts and some example generations). Unlike SynthBio, we have no reference biographies for these entities, so we devise a new evaluation procedure that tries to measure the factuality of the summaries. Specifically, we wrote questions for each property in the schema of the data, and then prompted GPT-3.5-4K to answer them using only the rendered text (see App. B for details). If the answer was not present in the text, we asked the model to answer with "Unknown." (the percentage of questions answered in this way is reported by the unknown rate; UR). We then computed both exact match accuracy and ROUGE of the provided responses against the response specified in the JSON; we include the latter as it is provides more leeway in the exact phrasing of the response than the exact match metric. Similar to §3.1, we do not report indirect SymGen results since it does not function properly using GPT-3.5-4K.

**Results.** The counterfactual obituary results are shown in Tab. 2. We find that in the zero-shot case, SymGen slightly outperforms our baseline, whereas in the few-shot case we find that it slightly underperforms the baseline; regardless, these differences are small, which again indicates that SymGen likely does not cause any meaningful degradation in the textual quality of the final generation. However, when we condition on whether the model gave a response versus stating it did not know the answer (Tab. 6), we find that performance is further improved in the zero-shot case and comparable in the few-shot setting. One possible explanation for this is that symbolic generation includes slightly less information than regular generation (especially in the few-shot case, as seen by the higher unknown rate), which in turn leads to more incorrect answers, since more questions were

left unanswered. That is, SymGen seems to favor precision at the expense of recall, which may be desirable in some applications.[5]

# 4 Verifiability Study

## 4.1 Assessing symbolic reference accuracy

We first aim to determine whether the fields attributed to different spans of text are correct given their context. This is important since, in principle, symbolic references may be incorrect even when the final text is accurate; for example if two fields in our data have the same value, but a symbolic reference is made to the wrong field, then the rendered text would be correct but not the reference.

**Metric.** For a given generation $\tilde{y}$, we measure the *accuracy* of the generated symbolic references. The accuracy $P$ is defined by comparing whether a generated field $\hat{f}_j \in \mathcal{F}$ is the same as the intended $f_j \in \mathcal{F}$ given the context: $\frac{1}{m} \sum_j \mathbb{1}[\hat{f}_j = f_j]$, where $m$ is the number of symbolic references in a given generation.

**Regex baseline.** Besides SymGen, we also use a simple regular expression-based baseline to generate symbolic references. In short, for each field $f_i$ in the data, we search for its corresponding value $v_i$ in the text and attribute any of its occurrences to $f_i$.

**Evaluation.** We sample 20 test examples from the Rotowire (Wiseman et al., 2017) dataset, which is a challenging testbed for this study due to the lengthy data associated with every generation. For each example, we compare the direct SymGen generation to the regex baseline, using either GPT-3.5-16K or GPT-4-32K as the base LLM. Three of the authors of this paper independently annotated the symbolic references in each example according to whether it was judged to be contextually correct or not, resulting in in 9068 annotations in total, with an inter-annotator agreement of 97.8%. Overall, we find that when the accuracy of SymGen is 99.77% and 99.52% with GPT-3.5-16K and GPT-4-32K as the base LLMs, respectively, whereas the one the baseline is 35.40% and 46.10%. This highlights that not only SymGen is extremely precise when generating symbolic references, but also that such accuracy cannot be obtained with simple, post-hoc methods.

## 4.2 Human evaluation of improved verifiability

Having established that textual quality is mostly unaffected by SymGen (§3) and that the symbolic references are correct and non-trivial (§4.1), we now turn to whether our symbolically grounded generations actually aid in verifiability. To do so, we conduct a human study to ascertain whether the annotations enabled by SymGen actually aid users in verifying LLM generations. We once again focus on Rotowire since it is a challenging testbed for verification: summaries contain many numbers referring to a wide range of fields, and slight errors might be hard to spot.

**Annotation data.** We picked five SymGen-generated (Indirect, GPT-4-32K) summaries of different games in the Rotowire dataset and manually verified them for correctness. We then generated three version of each document that were inconsistent with the data by picking a random symbolic reference and replacing it with another symbolic reference with a different value that was within 2 units of the original value of the field. We prevent the resulting value from being nonsensical, e.g., we restrict percentages to be within 0 and 100. For each of these four documents, we created two versions: a version with annotations, and one without any annotations (i.e., as if it were generated by a regular LLM).

---

[5]Here we characterize the "precision" in a loose sense and leave the formal evaluation of precision/recall for future work.

|  |  | 0-shot | 3-shot |
|---|---|---|---|
| GPT-3.5-16K | Baseline | 93.75 | 90.63 |
|  | Direct | 65.63 | 68.75 |
| GPT-4-32K | Baseline | 93.75 | 87.50 |
|  | Direct | 87.50 | 87.50 |
|  | Indirect | 90.63 | 93.75 |

(a) Results on the financial QA setting.

|  | GSM8K | | GSM-hard | |
|---|---|---|---|---|
|  | GPT-4 | GPT-3.5 | GPT-4 | GPT-3.5 |
| CoT | 95.0 | 81.0 | 64.0 | 53.5 |
| PAL | 95.0 | 82.0 | 79.5 | 73.0 |
| SymGen | 95.0 | 79.0 | 75.0 | 60.5 |

(b) Results on GSM8K.

Table 3: For financial QA, we report the acceptability of the answers in the zero- and 3-shot settings. Refer to App. C for details. For GSM8K, we compare SymGen reasoning with other reasoning methods on GSM8K and GSM-hard; all approaches were evaluated using GPT-4-8K. Refer to §5.2 for a discussion.

**Survey overview.** We ask annotators to annotate four summaries: two of them contain SymGen annotations, allowing them to inspect the provenance of different numbers in the text when hovering over a number using their cursor, and two containing no annotations (i.e., the output of a regular LLM). Each summary is equally likely to come from a pool of correct or incorrect summaries, and the first step of each annotation is for the participant to determine whether the summary is correct (i.e., there are no inconsistencies between the summary and the source table) or incorrect (i.e., there is an inconsistency between the summary and the table). Crucially, this first step is timed. Answering this question reveals two more questions, asking the annotators to rate their confidence in their answer, and how easy it was for them to reach a decision, on a Likert scale. We ask these questions in order to measure whether SymGen annotations meaningfully impact (i) human accuracy at finding errors and (ii) how easy it is to verify a summary, where both *objective* ease (i.e., how long did it take to reach a decision) and *subjective* ease (i.e., how hard did it *feel* to go through and verify the text). At the end of the study, annotators are provided an exit survey were they are asked to rate whether they prefer annotations (over no annotations) with respect to both (i) the confidence in their decisions and (ii) their ease in reaching a decision. We also reward annotators for each question answered correctly to ensure that they are spending adequate effort on the task. More details on the study design, exact questions, and interface are given in App. F.

**Enrollment.** We recruited 60 annotators for the study via Prolific. See App. F for more details, including selection criteria, quality filters, etc.

**Results.** We find that annotators were equally likely to identify errors regardless of whether SymGen annotations were shown. However, when annotations were displayed, they took on average 20% less time to reach a decision about whether an error was present or not. This makes sense, since the annotations do not change whether the text is correct or not, they should only facilitate the process of verifying it. We further find that annotators feel 5% more confident of their final answers when using our approach and perceive the verification task to be 14% easier. Finally, from exit survey, we have that 71.67% of participants agree that annotations made them more confident in their answers, and 83.33% agree that annotations made the verification task easier. In all, the results suggest that SymGen annotations enable a faster and more pleasant verification experience, with some annotators reporting "The annotations help tremendously", "[t]he annotations were nice and definitely helped", and "[...] I could still compare the data without annotations it just took longer to find each piece of data". For more details, refer to Tab. 7 in App. H.

## 5 Extensions to other use-cases

In the preceding sections, we established that the textual quality of SymGen is comparable to a regular prompted LLM (§3), that its annotations are accurate and non-trivial (§4.1) and that SymGen indeed aids in verifiability (§4.2). Now, we turn to some extensions of SymGen to question–answering (QA) and mathematical reasoning use-cases.

## 5.1 Question answering over structured data

Another possible application of symbolically grounded generation is to enhance verifiability in QA over structured data. To evaluate this setting, we construct a dataset of 32 finance-related questions about particular companies (e.g., "How does the book value of NFLX compare to that of ASML?"), coupled with structured company information. The results in Tab. 3a suggest that with GPT-4, all models offer reasonably comparable text quality across generation strategies, though SymGen underperforms when using GPT-3.5. Refer to App. C for more details.

## 5.2 Mathematical reasoning

Besides providing symbolic references to fields in the source data, SymGen can also be used to express *symbolic computations* over the variables, e.g., computing the half-time points of a basketball game based on the first two quarter scores (Fig. 2). By chaining a series of such operations, it opens up the possibility of interleaving arithmetic operations within language-based chain-of-thought reasoning. We explore whether this new capability comes without an overall textual quality penalty.

**Experimental setup.** Fig. 3 illustrates one approach for performing reasoning via symbolic generation for a math problem in GSM8K (Cobbe et al., 2021). Each generated computation step is coupled with an assignment statement in Jinja, which relates the natural language explanation of the computation with a symbolic expression. Compared to chain-of-thought reasoning (CoT; Wei et al., 2022), explicit use of symbolic computations should lead to more easily verifiable computational results of each step; compared to program-aided language models (PAL; Gao et al., 2022) and program of thoughts prompting (PoT; Chen et al., 2023), which recasts problems into programs and executes it to obtain a response, SymGen relies more on natural language as a scaffold, embedding symbolic computation within regular text.

**Dataset.** We compare the direct SymGen strategy against CoT and PAL on two datasets: GSM8K (Cobbe et al., 2021) and GSM-hard (Gao et al., 2022). GSM8K are grade school math problems like the one illustrated in Fig. 3 that typically require multiple steps of reasoning to solve. To create a more challenging testbed for LLMs, Gao et al. (2022) construct GSM-hard by replacing, for each problem in GSM8K, a randomly selected number with a large random number of up to seven digits. We use a random subset of 200 problems in the test set of GSM8K and GSM-hard for evaluation, as recent work find that it yields similar results to testing on the full 1.3K test set (Wang et al., 2023b).

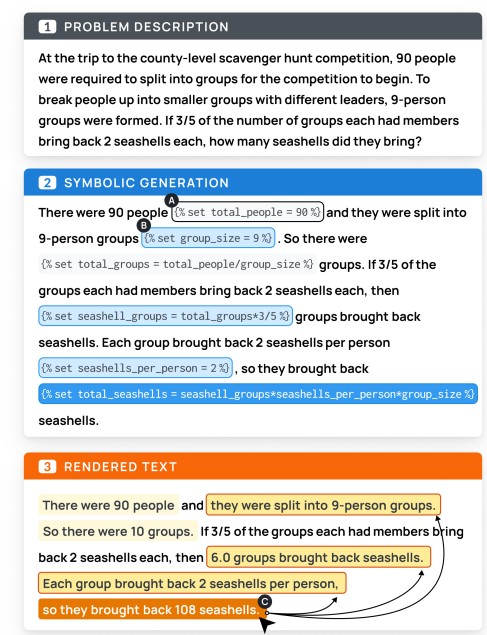

Figure 3: Illustration of SymGen reasoning on GSM8K, on an example generated by GPT-4. Given a math question (1), the LLM answers via direct symbolic generation, creating variables as needed (2), which can be rendered in a user-friendly manner (3). The syntax we use (Jinja) allows the creation of variables based on the source text (e.g., setting `total_people` to 90, **A**), and performing computation by referencing existing variables (e.g., calculating `total_groups` based on `total_people` and `group_size`, **B**). We can moreover explain how a computation step relates to previous ones (**C**).

**Results.** Tab. 3b compares the performance on GSM8K and GSM-hard using CoT, PAL, and our approach with GPT-3.5-4K and GPT-4-8K. Notably, it seems symbolic reasoning might not be necessary for GSM8K, since both our method and PAL perform similarly to CoT. However, when the computations are more complex as in GSM-hard, symbolic generation outperforms CoT by a clear margin when using both GPT-4-8K and GPT-3.5-4K. Our method slightly underperforms PAL on GPT-4-8K, with the margin being larger using GPT-3.5-4K. This is somewhat expected, given that PAL is explicitly geared towards solving such arithmetic problems and moreover makes use of richer programming elements. Inspecting our method's failures on this dataset, we found that many errors arose from sensible (but incorrect) attempts to reason symbolically about a math problem using Jinja notation. See App. G for an error analysis.

## 6 Related work

**Data-to-text generation.** Classic approaches for the data-to-text problem used hand-crafted templates to generate output sentences that interleave text with placeholder tokens which refer to relevant data fields (Kukich, 1983; McKeown, 1992; McRoy et al., 2000). Carefully engineered templates can provide strict guarantees against hallucinations, but require much manual work to specify, are domain-specific, and may sound robotic.

Data-driven methods for data-to-text initially focused on inducing templates from data Liang et al. (2009); Angeli et al. (2010); Howald et al. (2013); Konstas & Lapata (2013) which could be pipelined with a separate generation module. More recent approaches take advantage of large-scale datasets Wiseman et al. (2017); Parikh et al. (2020) to learn end-to-end neural networks Wang (2019); Li & Wan (2018); Yin & Wan (2022). (See Sharma et al. (2022) for a recent survey). Of particular note are methods that use template-like structures within neural generative modules to separate out the "what to say" from the "how to say it" Wiseman et al. (2018); Ye et al. (2020); Li & Rush (2020); Wang et al. (2021); Puduppully et al. (2022); Zhang et al. (2022). These methods typically still require a sizable training set, in contrast to the zero- and few-shot settings considered in the present work.

**Tool-augmented LLMs.** Recent work has augmented LLMs with tools/APIs to enhance their capabilities (Schick et al., 2023; Paranjape et al., 2023; Liang et al., 2023; Shen et al., 2023; Khot et al., 2023; Qin et al., 2023; Lu et al., 2023). These methods enable LLMs to generate string representations of API calls whose results are fed back in as additional context. Our method can be viewed as using a templating language as a tool. Whereas this line of work is usually concerned with enhancing LLM capabilities via external modules, our work instead focuses on using structured data as "tools" to improve verifiability. Our use of arithmetic computations within generations is also closely related to works that generate code snippets along with text to combine the benefits of symbolic computations and language-based reasoning (Gao et al., 2022; Chen et al., 2023; Zhang et al., 2023).

**LLM verification.** LLM verification is an orthogonal improvement to the reliability of LLMs, where the output of an LLM is automatically verified. Yue et al. (2023) and Gao et al. (2023) take an LLM generated response and a reference document it cites and use a separate module to determine whether the output of the LLM is entailed from the reference document. Verify-and-edit Zhao et al. (2023) uses self-consistency checks to discover potential mistakes, and corrects those mistakes with knowledge retrieval. Poesia et al. (2023) allow an LLM to call a logical backend as a tool, so that it can perform logically sound and certifiable reasoning. To the best of our knowledge, our work is among the first to focus on how LLM generations can be made easier to verify *by users* in data-to-text settings.

## 7 Conclusion

In this paper, we propose symbolically grounded generation (SymGen) as a simple approach to text generation that yields more verifiable text, by effectively linking portions of the output to the provided input data. Our evaluation suggests that symbolic generation increases verifiability without degrading quality.

## Acknowledgements

We thank AlphaVantage for providing us a key for academic use. This study was supported by funds from Liberty Mutual through the MIT Quest for Intelligence initiative. We thank Lauren Yu, Han Guo, Linlu Qiu, and Abbas Zeitoun for their helpful feedback.

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

## A    Discussion & Limitations

One limitation of SymGen is that it cannot completely ensure the avoidance of hallucinations. Although it is capable of generating accurate symbolic references, the surrounding context text may still be incorrect, thus leading to verifiable but incorrect generations. Our current studies primarily focuses on close-sourced models; one future work is to extend our method to open-sourced models and even finetune models for templated generation, using the dataset generated by SymGen (to the best of our knowledge, currently there are not available training data for this task).

## B    Obituary data evaluation

**Properties and questions.**    Tab. 4 lists the Wikidata properties we include in our entities alongside the questions we corresponding question we provide the LLM in the QA evaluation.

**Evaluation prompt.**    Refer to App. I.3 for the prompt we use in our QA evaluation of generated obituaries.

| Property ID | Property name | QA question |
| --- | --- | --- |
| P19 | place of birth | Where was X born? |
| P20 | place of death | Where did X die? |
| P27 | country of citizenship | What was X's country of citizenship? |
| P106 | occupation | What was X's occupation? |
| P1412 | languages spoken | What languages did X know? |
| P26 | spouse | Who was X's spouse? |
| P22 | father | Who was X's father? |
| P25 | mother | Who was X's mother? |
| P39 | position held | What position did X hold? |
| P166 | award received | What awards did X receive? |
| P140 | religion or worldview | What was X's religion? |
| P69 | educated at | Where did X study? |
| P119 | place of burial | Where is X buried? |
| P463 | member of | What was X a member of? |
| P509 | cause of death | What was the cause of X's death? |
| P101 | field of work | What was X's field of work? |
| P800 | notable work | What was X's notable work? |
| P1344 | participant in | What was X a participant in? |
| P108 | employer | Who was X's employer? |
| P1066 | student of | X was an student of whom? |
| P802 | student | Who was X's student? |
| P184 | doctoral advisor | Who was X's doctoral advisor? |
| P185 | doctoral student | Who was X's doctoral student? |
| P1411 | nominated for | What was X nominated for? |
| P551 | residence | Where did X live? |
| P512 | academic degree | What was X's academic degree? |

Table 4: Wikidata properties and corresponding QA question for the obituary dataset. We use X as a placeholder for the entity's name.

## C Financial question answering over structured data

Another possible application of symbolically grounded generation is to enhance verifiability in question-answering (QA) over structured data. This is especially motivated by recent work on tool-augmented LLMs (Schick et al., 2023; Karpas et al., 2022), which are able to condition their generation on the results of API calls, which are often given in a structured format.

**Dataset.**  To evaluate this setting, we construct a dataset of 32 finance-related questions about particular companies (e.g., "How does the book value of NFLX compare to that of ASML?"), coupled with company information from the AlphaVantage company overview API endpoint.[6] Details about the questions and postprocessing are shown in App. D.

**Experimental setup.**  We prompt the LLMs to answer finance-related questions given some company information. To evaluate the answers, we had one of the authors annotate each answer as acceptable or not; in general, an answer was deemed acceptable if the final answer was correct (see App. D for more details). We considered two LLMs: GPT-3.5-16K and GPT-4-32K, due to the length of the JSON data. Our prompts and some example outputs are given in App. I.3.1; we used three-shot prompting for the in-context learning setting.

**Results.**  For the financial dataset, we report the percentage of responses judged as acceptable under each approach in Tab. 3a. The results suggest that GPT-3.5-16K is unsuited for symbolic generation on this task, falling substantially short of the baseline. On the other hand, for GPT-4 all models offer reasonably comparable performance across generation strategies. See App. G for a qualitative analysis of some errors.

## D Financial question answering human evaluation

We had a total of 32 questions in the financial QA evaluation; the questions are shown in Tab. 5. There are two types of question: pairwise-comparison questions (`pair`) and multiway-comparison questions (`multi`). For the pairwise comparison questions, we only include information from the two companies in question. For the multiway-comparison questions we show data from seven tickers: ORCL, ASML, TSLA, GOOG, CRM, AVGO, NFLX.

The specific structured input provided is the result the company overview endpoint from the AlphaVantage API.[7] We applied some minor postprocessing to the API response to ensure that all fields that are numerical in nature (e.g., `SharesOutstanding`, `MarketCapitalization`) are actually numerical (i.e., integers or floats) in the JSON as opposed to strings. Note that all questions are answerable from the data provided data (e.g., there is a field `MarketCapitalization` in the structured data that allows the `pair_market-cap` questions to be answered).

To evaluate the generations, we had one of the authors in the paper annotate each of the responses according to whether the answer was acceptable or not. We adopted a loose criteria for acceptability:

1. If the question had an objective answer (e.g., the answer to "Which company has a largest market capitalization, GOOG os ASML?" is ultimately either GOOG or ASML) then the generation was acceptable as long as the final answer given by the question was acceptable. Note that if the *reasoning* given for the answer was incorrect but the final answer was correct, the answer was still treated as acceptable. This choice is deliberate, since in certain cases it can be hard to gauge whether

---

[6]https://www.alphavantage.co/documentation/#company-overview

[7]An example is shown here: https://www.alphavantage.co/query?function=OVERVIEW&symbol=IBM&apikey=demo.

| Question code | Question |
|---|---|
| pair_market-cap | Which company has the largest market capitalization, GOOG or ASML? |
| pair_market-cap | Which company has the largest market capitalization, ASML or ORCL? |
| pair_market-cap | Which company has the largest market capitalization, TSLA or CRM? |
| pair_pe-ratio | How do the P/E ratios of NFLX and ASML compare? |
| pair_pe-ratio | How do the P/E ratios of GOOG and CRM compare? |
| pair_pe-ratio | How do the P/E ratios of NFLX and AVGO compare? |
| multi_dividend | Which company has a higher dividend yield? |
| multi_revenue-mult | Which company has a higher revenue multiple? |
| pair_book-value | How does the book value of NFLX compare to that of ASML? |
| pair_book-value | How does the book value of ORCL compare to that of NFLX? |
| pair_book-value | How does the book value of CRM compare to that of TSLA? |
| multi_growth-yoy | Which company grew the fastest YoY in the most recent quarter? |
| pair_operating-margin | How do the operating margins of GOOG and CRM compare? |
| pair_operating-margin | How do the operating margins of NFLX and CRM compare? |
| pair_operating-margin | How do the operating margins of AVGO and TSLA compare? |
| multi_return-assets | Which company has a higher return on assets (TTM)? |
| pair_eps | How does the EPS (earnings per share) of ASML compare to AVGO? |
| pair_eps | How does the EPS (earnings per share) of TSLA compare to ORCL? |
| pair_eps | How does the EPS (earnings per share) of CRM compare to GOOG? |
| pair_beta | What are the respective market betas for CRM and GOOG? |
| pair_beta | What are the respective market betas for GOOG and NFLX? |
| pair_beta | What are the respective market betas for NFLX and AVGO? |
| pair_50d-avg | How do the 50-d moving averages between AVGO and GOOG compare? |
| pair_50d-avg | How do the 50-d moving averages between TSLA and ORCL compare? |
| pair_50d-avg | How do the 50-d moving averages between ORCL and GOOG compare? |
| pair_forward-pe-ratio | What's the difference in the forward P/E ratio between ASML and NFLX? |
| pair_forward-pe-ratio | What's the difference in the forward P/E ratio between AVGO and ASML? |
| pair_forward-pe-ratio | What's the difference in the forward P/E ratio between CRM and NFLX? |
| multi_quart-growth | Which company has a stronger quarterly earnings growth (YoY)? |
| pair_ebitda | How do ORCL and GOOG's EBITDA multiples compare? |
| pair_ebitda | How do ASML and TSLA's EBITDA multiples compare? |
| pair_ebitda | How do ASML and GOOG's EBITDA multiples compare? |

Table 5: The 32 questions that were considered for the financial QA evaluation.

|        |          | C-EM (%) | C-ROU.-1 | C-ROU.-2 | C-ROU.-L |
|--------|----------|----------|----------|----------|----------|
| 0-shot | Baseline | 80.35    | 87.56    | 62.55    | 87.50    |
|        | Direct   | 84.07    | 89.59    | 63.09    | 89.57    |
| 2-shot | Baseline | 83.95    | 88.85    | 63.33    | 88.85    |
|        | Direct   | 84.28    | 88.88    | 62.43    | 88.88    |

Table 6: GPT-3.5-4K results of the automated question–answering evaluation on the obituary dataset, when we condition of the model not having abstained from giving an answer, i.e., when the answer given by the model was not "Unknown." We report the conditioned exact match accuracy (C-EM (%)) and ROUGE (C-ROU.-1, C-ROU.-2, C-ROU.-L; Lin, 2004) of the inferred answers from the generated text against the corresponding true answers given by the source data. Refer to §3.2 for a discussion.

    the reasoning is correct despite the answer being correct. To reduce subjectivity as much as possible, we adopted the loosest possible interpretation of acceptability for questions with objective answers.

2. If the answer did not have an objective answer (e.g., there are multiple sensible answers to "How do ORCL and GOOG's EBIDTA multiples compare?"), then we deem an answer acceptable if the reasoning in the answer is sensible and correct.

3. If the model failed to generate an answer (e.g., rendering error, or did not answer the question) then the answer was not acceptable.

## E  Conditioned obituary results

In Tab. 6, we include the results on the obituary dataset when we condition on the QA model not having abstained from answering.

## F  Details on the human evaluation of improved verifiability

We include further details on the human study below.

### F.1  Detailed breakdown of summary steps

The following are the different pages the annotators see:

1. **Instruction page**: They first see a page that gives them an overview of the study and the interface, a description of the questions they will answer, the structure of the study (with the approximate times to complete each section, alongside information about compensation), and a section with further notes. We make it clear here that they are being timed, and that the first three annotations will be practice examples and the next four are the actual examples they need to annotate. For simplicity, we use System A to refer to generations without annotations, and System B to refer to the system with annotations.

2. **Practice round 1**: Gives an example of a document without annotations and without errors. Specifically, we say: "The following example, which could have been generated by system A, contains nothing that directly contradicts the table on the left (we have checked it ourselves by going through the different numbers mentioned in the text, but you will have to do it yourself for future examples). For this reason, you should say that the text is consistent with the source data below (this is an attention check). For these practice questions, it doesn't matter what you write down for confidence and ease, but after the practice rounds are finished, you should start answering those."

3. **Practice round 2**: Gives an example document without annotation but with an error. Specifically, we say: "The following is an example, that also could have been

generated by System A, is identical except that it now contains an error. Specifically, the text says that number of rebounds by Andrew Bogut made was 7, but the number in the table says it is actually 6 (you have to scroll down the table to see Andrew Boguts's statistics). For this reason, you should state that the text is inconsistent with the source data below (this is an attention check). Again, since this is a practice question, it doesn't matter what you answer for your confidence and ease of finding the answer."

4. **Practice round 3**: Gives an example of a document with annotations, and with an error. Specifically, we say: "The example shown below, which illustrates system B's annotations, is very similar but also contains an error: The text says that the number of rebounds that Stephen Curry made 38 points, but in reality he scored 37 points. If you hover your cursor over the number 38, a tooltip will show up, showing that the 38 actually refers to the Golden State Warrior's 3Pt Field Goal Percentage. For this reason, you should state that the text is inconsistent with the source data below (this is an attention check). Again, since this is a practice run, it doesn't matter what you answer for your confidence and ease of finding the answer, but you should start answering them properly from now on."

5. **Annotation 1**: First example they need to annotate.

6. **Annotation 2**: Second example they need to annotate.

7. **Annotation 3**: Third example they need to annotate.

8. **Annotation 4**: Fourth example they need to annotate.

**Quality checks.** We request a participant return their submission if they get two of the practice rounds wrong. In our opinion, our instructions for those rounds are clear as to what they should do (i.e., they give the correct answer), so participants that are reading the instructions should get these correct. If they get only one wrong, we allow them to complete the study and compensate them as other participants, but do not use their results in our analysis.

**Sampling of annotations.** When the user passes the practice round, we show them four documents to annotate. Our sampling process obeys the following properties:

1. Each of the underlying documents is unique (i.e., we pick 4 out of the 5 documents to show the annotator)

2. Exactly 2 of the documents are shown with annotations, and two are shown without annotations. These may appear in any order.

3. Each document has a 50% chance of being consistent with the text. If a document is sampled to be inconsistent, then we sample one of the three possible incorrect variants to show, uniformly at random.

### F.2 Questions asked to annotators

For each document, the annotators are asked:

1. Is the text consistent or inconsistent with the source data?
   - Possible answers: Consistent or Inconsistent
   - Note: Answering this question stops a timer that begins running when the page is loaded. This is the annotation time we report.

2. How confident are you in your answer? [1 means least confidence (completely unsure), 5 means most confidence (absolutely certain)]
   - Possible answers: 1, 2, 3, 4, 5

3. How easy was it reach your decision? [1 is hardest, 5 is easiest]
   - Possible answers: 1, 2, 3, 4, 5

After annotating all documents, the exit survey asked them "Please rate the degree to which you agree with the following statement:"

1. System B (i.e., with annotations) gave me more confidence in my answers that System A (i.e., no annotations).
   - Possible answers: Strongly disagree, Disagree, Neutral, Agree, Strongly agree
2. System B (i.e., with annotations) made it easier for me to do the task than System A (i.e., no annotations).
   - Possible answers: Strongly disagree, Disagree, Neutral, Agree, Strongly agree

And finally there we added "Optionally, you can leave some comments about the study below." alongside a textbox for comments.

### F.3 Interface

The following images illustrate the annotation interface:

1. Fig. 4 depicts the instruction page shown to each annotator.
2. Fig. 5 illustrates the full annotation interface for a document, after it has been filled out.
3. Fig. 6 illustrates the the exit survey, after it has been filled out.
4. Fig. 7 illustrates how the interface changes when you hover over an annotation with your cursor. Specifically, a tooltip appears (describing the provenance for the field) and the respective field gets highlighted on the left. The tooltips and the table are both human readable (the values are manually written based on the schema of the data, which is known).

## G Qualitative error analysis

Fig. 8 shows some example errors in SymGen generations. Below we include further qualitative comments on the errors produced by the systems we analysed.

**GSM error anlaysis.** We inspected GPT-3.5-4K failures by our model. In one case the model generates `{% set initial_marbles = 'x' %}`, in an attempt to effectively create a variable, as one would do when reasoning about a problem algebraically. In another case, the statement `{% set red_paint = white_paint = purple_paint %}` was generated in response to "There are equal amounts of red, white, and purple paint in the mural," which is incorrect Jinja despite making sense conceptually.

**Financial error analysis.** During our analysis, we found that GPT-3.5-16K struggled with CoT-like (Wei et al., 2022) behavior in symbolic form, e.g., when it required comparing a set of numbers and determining the largest one, it would generate references to the numbers involved in the comparison (as is standard with CoT), but it would not determine the largest one correctly (e.g., Fig. 8C). We also found evidence of the models struggling with plausible but syntactically incorrect Jinja generations. In fact, most errors in the symbolic form of the output were because of incorrectly referencing one particular variable, viz., `50DayMovingAverage` (e.g., Fig. 8D). This suggests that techniques for constrained decoding with LLMs (Shin et al., 2021; Scholak et al., 2021; Poesia et al., 2022; 2023; Wang et al., 2023a) could be employed to curtail syntax errors.

## H Human study detailed results

For the detailed results regarding the human study in §4.2, refer to Tab. 7.

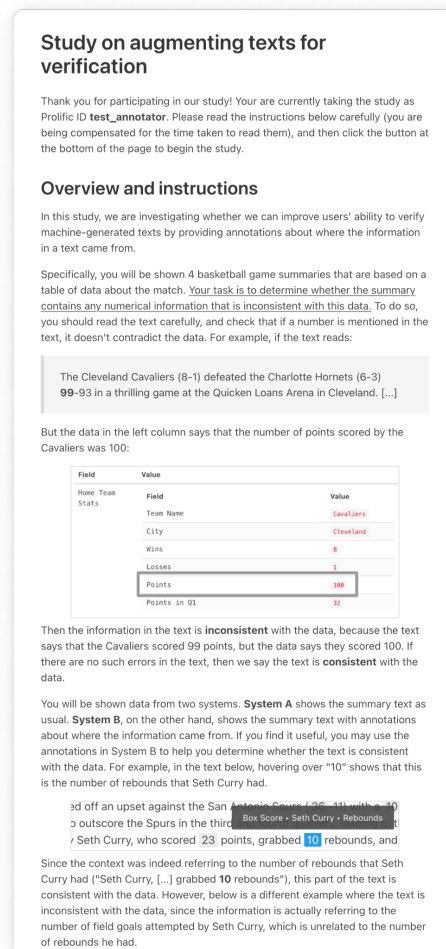 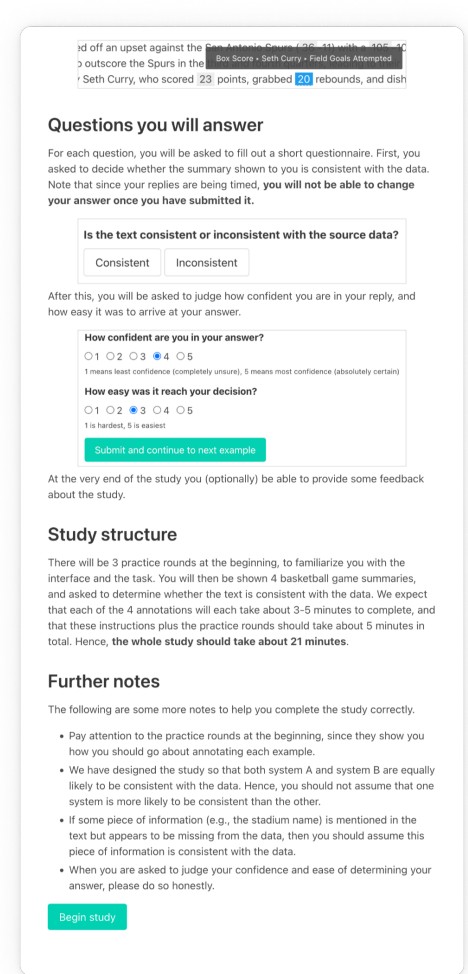

(a) The top half of the instruction page.  (b) The bottom half of the instruction page.

Figure 4: Instructions page shown to every annotator, before they begin the study.

|          | Acc.            | Time            | Conf.           | Effort          |
|----------|-----------------|-----------------|-----------------|-----------------|
| SymGen   | $80.83 \pm 3.61$ | $1.98 \pm 0.12$ | $4.63 \pm 0.07$ | $4.14 \pm 0.10$ |
| Baseline | $80.00 \pm 3.67$ | $2.50 \pm 0.16$ | $4.44 \pm 0.09$ | $3.63 \pm 0.11$ |

Table 7: Results of the human study, comparing a system with SymGen annotations to one without. We report accuracy (Acc.), time taken to determine if an example is correct in minutes (Time), and perceived confidence (Conf.) and effort (Effort) on a Likert scale (i.e., 1–5). For Conf. and Effort, higher scores indicate more confidence and less effort respectively. See §4.2 for details. Error bars correspond to standard error on the mean.

## I Experimental details

Unless otherwise specified, we run all experiments using the Azure API and use the model version `0613`. We state the specific version of the model we use in the main text. We use the SacreBLEU, ROUGE and BERTScore implementations in Huggingface evaluate.[8]

---

[8]https://github.com/huggingface/evaluate

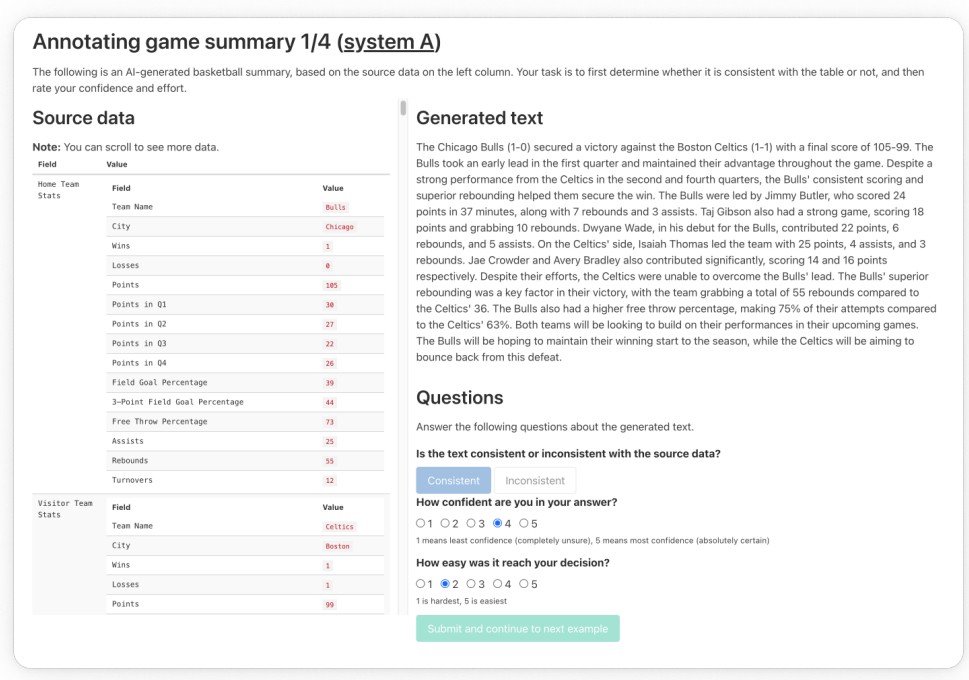

Figure 5: Annotation interface for a document, as it appears after answering all questions. The second and third questions only appear after the first one is answered, and the submit button is enabled only when everything has been answered.

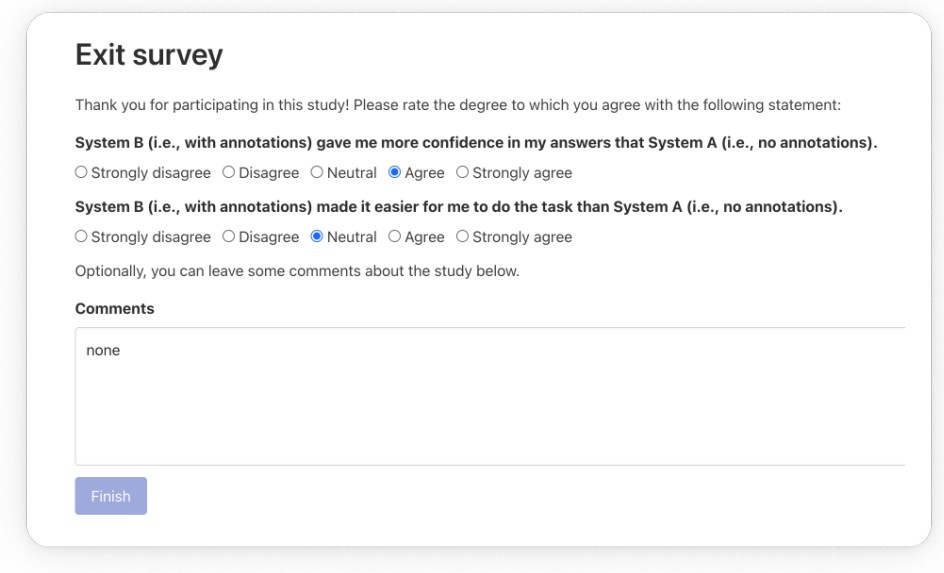

Figure 6: Exit survey at the end of the human study.

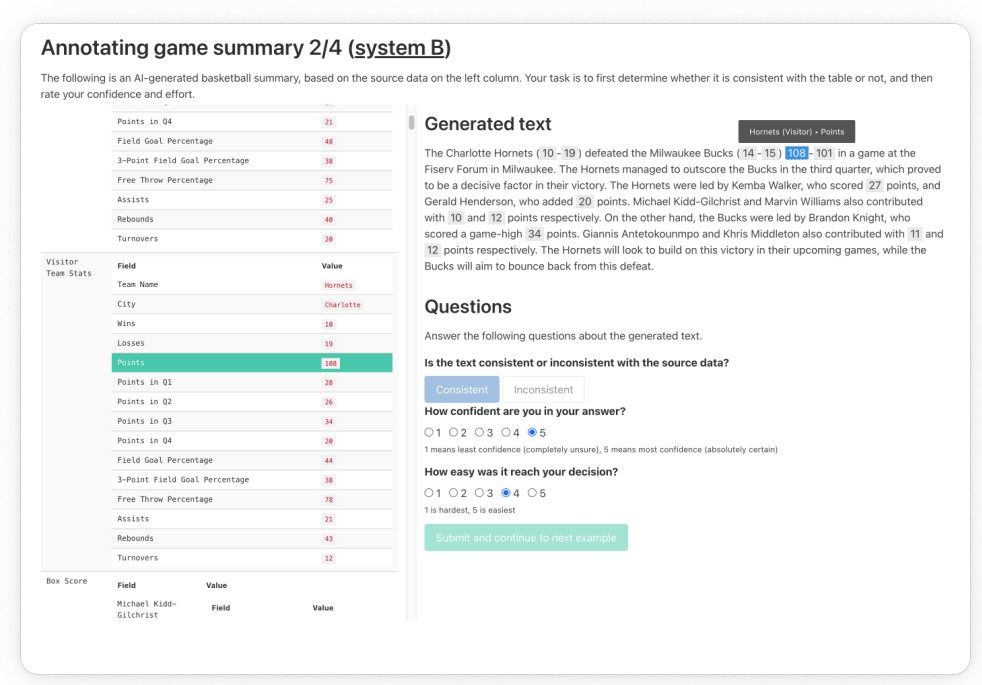

Figure 7: Effect of hovering over an annotation in a document that has annotations. A tooltip shows up containing a short description of the field, and the corresponding entry in the table is highlighted.

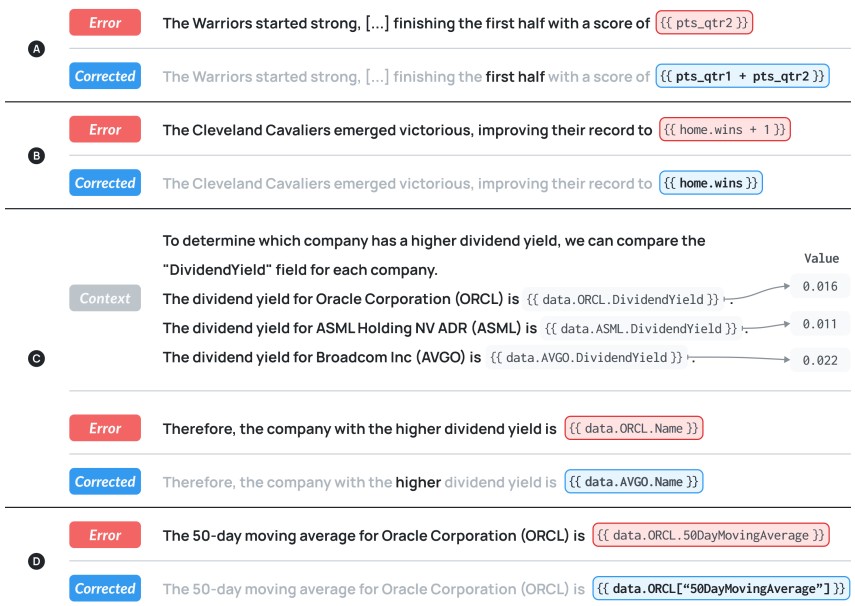

Figure 8: SymGen makes the identification of hallucinations easier. In **A** SymGen makes an error (indicated with dark red background) when the model needs to add up two values; in **B**, the model incorrectly increments the values by one. In **C** the model generates a correct chain-of-thought, but fails, likely because the symbolic references make it harder to check which value is actually highest. Finally, in **D**, we see a sensible but syntactically incorrect Jinja expression, which causes the whole generation to become unrenderable. Note how even when there are errors, symbolic generation provides a glimpse into the models' internal reasoning, making it easier to understand the motive for such errors (e.g., generating +1 in **B**).

**Licensing information.** The following resources that were used in our paper are licensed as follows (all are in English):

1. **SynthBio**: Apache 2.0
2. **Rotowire**: Unable to determine
3. **GSM**: MIT

## I.1 Datasets

Tab. 8 shows the statistics of the datasets we consider. We provide some more details about each dataset and its processing below.

**Rotowire.** We postprocess the JSON to convert it into a nested, key–value data structure that resembles the output of an API. We found this to be a necessary step, and found SymGen to not work well when most of the data is exposed as an array, likely because of implicit chain-of-thought-like behavior when the model specifies a key, e.g., `{{ data.box_score.shabazz_napier.fgm }}` should be easier for an LLM to reason about than `{{ data.box_scores[1][5] }}`.

**Counterfactual obituaries.** To obtain the entities, we scraped Wikidata[9] and collected all entities whose occupation is a subtype of scientist, additionally filtering for entities with 75 or more sitelinks to ensure relatively well-known scientists were selected. To construct

---

[9]https://www.wikidata.org/wiki/Wikidata:Main_Page

| Dataset | Domain | Eval # | Field types | Field # | Ref? |
|---------|--------|--------|-------------|---------|------|
| SynthBio | Biographies | 364 | Text | 18.96 | ✓ |
| Obituary | Obituaries | 168 | Text, Number | 20.98 | |
| Rotowire | Sports | 100 | Text, Number | 644.16 | ✓ |
| Financial | Financial | 32 | Text, Number | 127.94 | |
| GSM | Mathematics | 200 | None | N/A | ✓ |

Table 8: Comparison of the four datasets we consider in our study. We note the domain of the dataset (Domain), the number of datapoints we evaluate on (Eval #), the types of the fields of the data, the average number of fields in each example's data (Field #), and whether there is a reference generation for each datapoint in the dataset (Ref?).

the counterfactual variants of each entity, we take each entity's attribute and progressively corrupt a subset of its properties,[10] by sampling a different value from the empirical distribution of that property's values from the collected data. For each entity, we create copies that have 50% and 100% of the selected properties corrupted.

We then constructed a dictionary for each entity containing a value for each one of 27 manually selected Wikidata properties, e.g., country of citizenship, place of birth, etc. (see App. B for a complete listing). If an entity did not possess such a property in its Wikidata page, we do not add it to the entity's dictionary; if the entity possessed multiple values associated with that property, we selected the first one that was returned by the Wikidata API. This results in one dictionary of property–value attributes for each scientist. To construct the counterfactual variants of each entity, we take each entity's dictionary and progressively corrupt a subset of its properties,[11] by sampling a different value from the empirical distribution of that property's values from the collected data. For each entity, we create copies that have 50% and 100% of the selected properties corrupted. All of these are added back to obtain the final dataset.

## I.2 Prompts

We provide the prompts we use in the sections below.

### I.2.1 SynthBio prompts

---

**SynthBio prompt (Baseline)**

**System prompt:**
Your task is to write a biography about a person whose description is given in JSON. You must mention and be faithful about every property of the person in your biography.

---

**Few shot examples (optional):**

👤 **User:**
JSON:

```json
{
    "data": {
        "name": "Hanna Skold",
        "gender": "female",
        "nationality": "Swedish",
        "birth_date": "14 November 1856",
        "birth_place": "Gotlandso, Sweden",
```

---
[10]Other than name, age and gender.
[11]Other than name, age and gender.

```
            "death_date": "24 December 1935",
            "death_place": "Santa Cruz, California",
            "death_cause": "unknown",
            "start_age": "17",
            "notable_ascents": "Pico de Orizaba, Aconcgua, Grand
                Teton, Cerro Fitzroy",
            "final_ascent": "Mount Rainier",
            "partnerships": "Gertrude S. Cooper",
            "mother": "Karin Andersen",
            "children": "Lillian Skold"
        }
    }
    ```
```

**Assistant:**

Hanna Skold (born 14 November 1856) was a Swedish mountaineer. She was born in Gotlandso, Sweden. She started climbing at the age of 17. She climbed Pico de Orizaba, Aconcagua, Grand Teton, and Cerro Fitzroy. Skold made her final ascent on Mount Rainier. Skold was the daughter of Karin Andersen and her daughter was Lillian Skold. She collaborated with Gertrude S. Cooper. She died on December 24, 1935 in Santa Cruz, California.

*One more few-shot example like the one above (hidden to save space)*

**User:**

*Data for biography to be generated.*

EXAMPLE INPUT
JSON:

```json
{
    "data": {
        "name": "Helmut Kaiser",
        "gender": "male",
        "nationality": "German",
        "birth_date": "17 August 1863",
        "birth_place": "Nuremberg",
        "death_date": "October 17, 1947",
        "death_place": "Frankfurt am Main, Germany",
        "death_cause": "heart disease; artery sclerosis",
        "resting_place": "Frankfurt am Main, Germany",
        "alma_mater": "Philosophical University of Erlangen-Nuernberg, 1886",
        "education": "Ph.D. in Philosophy",
        "occupation": "philosopher, theologian;",
        "notable_works": "Philosophical Problems of Theism",
        "language": "Germany",
        "genre": "Philosophical Theology",
        "awards": "Honorary Member of the Lutheran Liturgical Institute at the University of
            Chicago; University Grant of City of Frankfurt am Main in 1911",
        "mother": "Berta Kaiser",
        "father": "Adolf Weber",
        "partner": "Frieda Schoenlein",
        "children": "Rudolf, Kurt, Gertraud, Marianne, Rosemarie"
    }
}
```

**Assistant:**

*LLM generates Synthbio summary without symbolic references.*

EXAMPLE OUTPUT (GPT-4-8K)

Helmut Kaiser was a renowned German philosopher and theologian, born on August 17, 1863, in Nuremberg, Germany. He completed his Ph.D. in Philosophy from the Philosophical University of Erlangen-Nuernberg in 1886. His most notable work was "Philosophical Problems of Theism". He was awarded the Honorary Member of the Lutheran Liturgical Institute at the University of Chicago and received the University Grant of the City of Frankfurt am Main in 1911. He was married to Frieda Schoenlein and they had five children: Rudolf, Kurt, Gertraud, Marianne, and Rosemarie. Kaiser passed away from heart disease and artery sclerosis on October 17, 1947, in Frankfurt am Main, Germany, where he was also laid to rest.

**SynthBio prompt (Direct)**

**System prompt:**
Your task is to write a biography about a person whose description is given in JSON. You must mention and be faithful about every property of the person in your biography. Instead of responding directly with the text, you must generate Jinja-style expressions in the text with references to fields in the provided JSON, e.g., given JSON that says:

```json
{
    "data": {
        "place_of_birth": "Ohio"
    }
}
```

then instead of generating "... was born in Ohio" you should generate "... was born in `{{ data.place_of_birth }}`". Note that you can only reference fields that exist in the JSON. The user will be displayed a version of this message that has the references filled in, and you should ALWAYS try to use such references instead of copying the text directly when possible.

---

**Few shot examples (optional):**

👤 **User:**
JSON:

```json
{
    "data": {
        "name": "Hanna Skold",
        "gender": "female",
        "nationality": "Swedish",
        "birth_date": "14 November 1856",
        "birth_place": "Gotlandso, Sweden",
        "death_date": "24 December 1935",
        "death_place": "Santa Cruz, California",
        "death_cause": "unknown",
        "start_age": "17",
        "notable_ascents": "Pico de Orizaba, Aconcgua, Grand
            Teton, Cerro Fitzroy",
        "final_ascent": "Mount Rainier",
        "partnerships": "Gertrude S. Cooper",
        "mother": "Karin Andersen",
        "children": "Lillian Skold"
    }
}
```

---

🖥️ **Assistant:**
`{{ data.name }}` (born `{{ data.birth_date }}`) was a `{{ data.nationality }}` mountaineer. She was born in `{{ data.birth_place }}`. She started climbing at the age of `{{ data.start_age }}`. She climbed `{{ data.notable_ascents.split(', ')[0] }}`, `{{ data.notable_ascents.split(', ')[1] }}`, `{{ data.notable_ascents.split(', ')[2] }}`, and `{{ data.notable_ascents.split(', ')[3] }}`. `{{ data.name }}` made her final ascent on `{{ data.final_ascent }}`. `{{ data.name }}` was the daughter of `{{ data.mother }}` and her daughter was `{{ data.children }}`. She collaborated with `{{ data.partnerships }}`. She died on `{{ data.death_date }}` in `{{ data.death_place }}`.

*One more few-shot example like the one above (hidden to save space)*

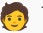 **User:**
*Data for biography to be generated*

EXAMPLE INPUT
JSON:

```json
{
    "data": {
        "name": "Helmut Kaiser",
        "gender": "male",
        "nationality": "German",
        "birth_date": "17 August 1863",
        "birth_place": "Nuremberg",
        "death_date": "October 17, 1947",
        "death_place": "Frankfurt am Main, Germany",
        "death_cause": "heart disease; artery sclerosis",
        "resting_place": "Frankfurt am Main, Germany",
        "alma_mater": "Philosophical University of Erlangen-Nuernberg, 1886",
        "education": "Ph.D. in Philosophy",
        "occupation": "philosopher, theologian;",
        "notable_works": "Philosophical Problems of Theism",
        "language": "Germany",
        "genre": "Philosophical Theology",
        "awards": "Honorary Member of the Lutheran Liturgical Institute at the University of
            Chicago; University Grant of City of Frankfurt am Main in 1911",
        "mother": "Berta Kaiser",
        "father": "Adolf Weber",
        "partner": "Frieda Schoenlein",
        "children": "Rudolf, Kurt, Gertraud, Marianne, Rosemarie"
    }
}
```

🖥 **Assistant:**
*LLM directly generates Synthbio summary with symbolic references.*

EXAMPLE OUTPUT (GPT-4)
{{ data.name }} was a renowned {{ data.nationality }} {{ data.occupation }} born on {{ data.birth_date }} in {{ data.birth_place }}. He was the son of {{ data.father }} and {{ data.mother }}. He studied at {{ data.alma_mater.split(', ')[0] }}, where he earned his {{ data.education }}. His most notable work was "{{ data.notable_works }}", written in his native language, {{ data.language }}. His contributions to {{ data.genre }} earned him recognition, including becoming an Honorary Member of the Lutheran Liturgical Institute at the University of Chicago and receiving the University Grant of City of Frankfurt am Main in 1911. He was partnered with {{ data.partner }} and they had five children: {{ data.children }}. {{ data.name }} passed away on {{ data.death_date }} in {{ data.death_place }} due to {{ data.death_cause }}. His final resting place is in {{ data.resting_place }}.

---

**SynthBio prompt (Indirect)**

**System prompt:**
Your task is to write a biography about a person whose description is given in JSON. You must mention and be faithful about every property of the person in your biography. After you generate the biography, you will be asked to rewrite it using Jinja-style expressions.

**Few shot examples (optional):**

👤 **User:**
JSON:

```json
{
    "data": {
        "name": "Hanna Skold",
        "gender": "female",
        "nationality": "Swedish",
        "birth_date": "14 November 1856",
        "birth_place": "Gotlandso, Sweden",
```

```
                "death_date": "24 December 1935",
                "death_place": "Santa Cruz, California",
                "death_cause": "unknown",
                "start_age": "17",
                "notable_ascents": "Pico de Orizaba, Aconcgua, Grand
                    Teton, Cerro Fitzroy",
                "final_ascent": "Mount Rainier",
                "partnerships": "Gertrude S. Cooper",
                "mother": "Karin Andersen",
                "children": "Lillian Skold"
        }
    }
```

🖥 **Assistant:**
Hanna Skold (born 14 November 1856) was a Swedish mountaineer. She was born in Gotlandso, Sweden. She started climbing at the age of 17. She climbed Pico de Orizaba, Aconcagua, Grand Teton, and Cerro Fitzroy. Skold made her final ascent on Mount Rainier. Skold was the daughter of Karin Andersen and her daughter was Lillian Skold. She collaborated with Gertrude S. Cooper. She died on December 24, 1935 in Santa Cruz, California.

👤 **User:**
Your job is to now rewrite the answer you provided above, but instead of responding directly with the text, you must generate Jinja-style expressions in the text with references to fields in the provided JSON, e.g., given JSON that says:

```json
{
    "data": {
        "place_of_birth": "Ohio"
    }
}
```

then instead of generating "... was born in Ohio" you should generate "... was born in {{ data.place_of_birth }}". Note that you can only reference fields that exist in the JSON. The user will be displayed a version of this message that has the references filled in, and you should ALWAYS try to use such references instead of copying the text directly when possible. Remember, you MUST make sure you ONLY use fields that exist in the JSON. You must NOT make up fields.

🖥 **Assistant:**
{{ data.name }} (born {{ data.birth_date }}) was a {{ data.nationality }} mountaineer. She was born in {{ data.birth_place }}. She started climbing at the age of {{ data.start_age }}. She climbed
{{ data.notable_ascents.split(', ')[0] }},
{{ data.notable_ascents.split(', ')[1] }},
{{ data.notable_ascents.split(', ')[2] }}, and
{{ data.notable_ascents.split(', ')[3] }}. {{ data.name }} made her final ascent on {{ data.final_ascent }}. {{ data.name }} was the daughter of
{{ data.mother }} and her daughter was {{ data.children }}. She collaborated with {{ data.partnerships }}. She died on {{ data.death_date }} in
{{ data.death_place }}.

*One more few-shot example like the one above (hidden to save space)*

👤 **User:**
*Data for biography to be generated*

EXAMPLE INPUT
JSON:

```json
{
    "data": {
        "name": "Helmut Kaiser",
        "gender": "male",
        "nationality": "German",
        "birth_date": "17 August 1863",
        "birth_place": "Nuremberg",
        "death_date": "October 17, 1947",
        "death_place": "Frankfurt am Main, Germany",
        "death_cause": "heart disease; artery sclerosis",
        "resting_place": "Frankfurt am Main, Germany",
        "alma_mater": "Philospohical University of Erlangen-Nuernberg, 1886",
        "education": "Ph.D. in Philosophy",
        "occupation": "philosopher, theologian;",
        "notable_works": "Philosophical Problems of Theism",
        "language": "Germany",
        "genre": "Philosophical Theology",
        "awards": "Honorary Member of the Lutheran Liturgical Institute at the University of
            Chicago; University Grant of City of Frankfurt am Main in 1911",
        "mother": "Berta Kaiser",
        "father": "Adolf Weber",
        "partner": "Frieda Schoenlein",
        "children": "Rudolf, Kurt, Gertraud, Marianne, Rosemarie"
    }
}
```

---

🖥️ **Assistant:**
*LLM generates SynthBio biography without symbolic references*

EXAMPLE OUTPUT (GPT-4-8K)
Helmut Kaiser was a German philosopher and theologian, born on August 17, 1863, in Nuremberg, to Adolf Weber and Berta Kaiser. He studied at the Philosophical University of Erlangen-Nuernberg, graduating in 1886 with a Ph.D. in Philosophy. His notable works include "Philosophical Problems of Theism". He wrote in German and his genre was Philosophical Theology. He received several awards, including becoming an Honorary Member of the Lutheran Liturgical Institute at the University of Chicago and receiving the University Grant of the City of Frankfurt am Main in 1911. He was partnered with Frieda Schoenlein and they had five children: Rudolf, Kurt, Gertraud, Marianne, and Rosemarie. Kaiser passed away on October 17, 1947, in Frankfurt am Main, Germany, due to heart disease and artery sclerosis. He was laid to rest in the same city.

---

🧑 **User:**
Your job is to now rewrite the answer you provided above, but instead of responding directly with the text, you must generate Jinja-style expressions in the text with references to fields in the provided JSON, e.g., given JSON that says:

```json
{
    "data": {
        "place_of_birth": "Ohio"
    }
}
```

then instead of generating "... was born in Ohio" you should generate "... was born in {{ data.place_of_birth }}". Note that you can only reference fields that exist in the JSON. The user will be displayed a version of this message that has the references filled in, and you should ALWAYS try to use such references instead of copying the text directly when possible. Remember, you MUST make sure you ONLY use fields that exist in the JSON. You must NOT make up fields.

---

🖥️ **Assistant:**
*LLM rewrites biography so it contains symbolic references.*

EXAMPLE OUTPUT (GPT-4-8K)

{{ data.name }} was a {{ data.nationality }} {{ data.occupation }}, born on {{ data.birth_date }}, in
{{ data.birth_place }}, to {{ data.father }} and {{ data.mother }}. He studied at the
{{ data.alma_mater.split(', ')[0] }}, graduating in {{ data.alma_mater.split(', ')[1] }} with a {{ data.education }}.
His notable works include "{{ data.notable_works }}". He wrote in {{ data.language }} and his genre was
{{ data.genre }}. He received several awards, including becoming an {{ data.awards.split('; ')[0] }} and receiving the
{{ data.awards.split('; ')[1] }}. He was partnered with {{ data.partner }} and they had five children:
{{ data.children }}. {{ data.name }} passed away on {{ data.death_date }}, in {{ data.death_place }}, due to
{{ data.death_cause }}. He was laid to rest in {{ data.resting_place }}.

## I.2.2 Rotowire prompts

### Rotowire prompt (Baseline)

**System prompt:**
Your task is to write a summary of a basketball game whose statistics are given in JSON.
You must remain faithful to the provided statistics in your summary.

**Few shot examples (optional):**

🧑 **User:**
JSON:

```json
{
    "data": {
        "home_line": {
            "name": "Magic",
            "city": "Orlando",
            "wins": 6,
            "losses": 9,
            "pts": 92,
            "pts_qtr1": 18,
            "pts_qtr2": 28,
            "pts_qtr3": 21,
            "pts_qtr4": 25,
            "fg_pct": 45,
            "fg3_pct": 19,
            "ft_pct": 70,
            "ast": 18,
            "reb": 44,
            "tov": 14
        },
        "visitor_line": {
            "name": "Heat",
            "city": "Miami",
            "wins": 7,
            "losses": 6,
            "pts": 99,
            "pts_qtr1": 22,
            "pts_qtr2": 27,
            "pts_qtr3": 19,
            "pts_qtr4": 31,
            "fg_pct": 52,
            "fg3_pct": 46,
            "ft_pct": 72,
            "ast": 17,
            "reb": 33,
            "tov": 16
        },
        "box_score": {
            "luol_deng": {
```

```
                        "player_name": "Luol Deng",
                        "first_name": "Luol",
                        "second_name": "Deng",
                        "city": "Miami",
                        "start_position": "F",
                        "min": 33,
                        "pts": 9,
                        "fgm": 4,
                        "fga": 9,
                        "fg_pct": 44,
                        "fg3m": 1,
                        "fg3a": 2,
                        "fg3_pct": 50,
                        "ftm": 0,
                        "fta": 0,
                        "ft_pct": 0,
                        "reb": 7,
                        "oreb": 1,
                        "dreb": 6,
                        "ast": 0,
                        "stl": 1,
                        "blk": 0,
                        "to": 2,
                        "pf": 3
                },
                ...
            }
        }
    }
```

Write a summary of the game described above.

**Assistant:**

The Miami Heat (7-6) took down the Orlando Magic (6-9) 99-92 on Saturday night at the Amway Center in Orlando. The Heat were just too much for the Magic when all was said and done, as they made a nice run in the fourth to get a seven-point win. What makes the win even more impressive is that Miami did it without guard Dwyane Wade (hamstring). The Heat got plenty of production from some key players to help get the job done though. They were led in scoring by Chris Bosh, who put up 32 points (13-20 FG, 4-6 3Pt, 2-2 FT) to go with 10 rebounds. Mario Chalmers had a huge night also, scoring 24 points (10-16 FG, 2-5 3Pt, 2-4 FT) to go with eight assists and five rebounds. Miami fans have to be happy to see rookie Shabazz Napier stepping up big, as he played extended minutes (28), and scored 12 points (4-8 FG, 2-5 3Pt, 2-4 FT) to go with two rebounds, two steals and one assist. Miami has a quick turnaround, as they take on the Charlotte Hornets at home Sunday night. The Magic just couldn't get enough production to outscore the Heat on this night, as only three players hit double digits, with Nikola Vucevic having a monster night, scoring 33 points (12-24 FG, 9-11 FT), with 17 rebounds. Vucevic was just dominate, and grabbed 11 offensive boards on the night. The other production came from Victor Oladipo, who had 14 points (5-11 FG, 1-4 3Pt, 3-5 FT), with four rebounds, two assist and two steals, as well as Evan Fournier, who scored 11 points (5-15 FG, 1-5 3PT) with six assists and six rebounds. The Magic get a night off now, before taking on the Cleveland Cavaliers on Monday.

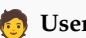 **User:**
*Data for game to be described*

EXAMPLE INPUT
JSON:

```json
{
    "data": {
        "home_line": {
            "name": "Hawks",
            "city": "Atlanta",
            "wins": 46,
            "losses": 12,
            "pts": 95,
            "pts_qtr1": 28,
            "pts_qtr2": 14,
            "pts_qtr3": 34,
            "pts_qtr4": 19,
            "fg_pct": 44,
            "fg3_pct": 45,
            "ft_pct": 75,
            "ast": 27,
            "reb": 42,
            "tov": 18
        },
        "visitor_line": {
            "name": "Magic",
            "city": "Orlando",
            "wins": 19,
            "losses": 41,
            "pts": 88,
            "pts_qtr1": 16,
            "pts_qtr2": 29,
            "pts_qtr3": 22,
            "pts_qtr4": 21,
            "fg_pct": 45,
            "fg3_pct": 35,
            "ft_pct": 54,
            "ast": 22,
            "reb": 40,
            "tov": 17
        },
        "box_score": {
            "tobias_harris": {
                "player_name": "Tobias Harris",
                "first_name": "Tobias",
                "second_name": "Harris",
                "city": "Orlando",
                "start_position": "F",
                "min": 37,
                "pts": 15,
                "fgm": 5,
                "fga": 9,
                "fg_pct": 56,
                "fg3m": 2,
                "fg3a": 4,
                "fg3_pct": 50,
                "ftm": 3,
                "fta": 4,
                "ft_pct": 75,
                "reb": 4,
                "oreb": 0,
                "dreb": 4,
                "ast": 1,
                "stl": 2,
                "blk": 1,
                "to": 2,
                "pf": 0
            },
            ...
        }
    }
}
```

Write a summary of the game described above.

🖥️ **Assistant:**
*LLM generates game summary without symbolic references*

The Atlanta Hawks (46-12) secured a victory against the Orlando Magic (19-41) with a final score of 95-88. The Hawks started strong with a 28-point first quarter and maintained their lead throughout the game, despite a slight dip in the second quarter. They managed to shoot 44% from the field and 45% from the three-point line. The Hawks also outrebounded the Magic 42-40 and had more assists (27-22).

The Hawks' top performers were Paul Millsap with 20 points, Al Horford with 17 points and 13 rebounds, and DeMarre Carroll with 15 points. Jeff Teague also contributed significantly with 17 points and 7 assists.

On the other side, the Magic struggled with their free throws, making only 54% of them. However, they managed to shoot 45% from the field and 35% from beyond the arc. The Magic's standout player was Nikola Vucevic, who scored 21 points and grabbed 15 rebounds. Victor Oladipo also had a solid performance with 19 points, while Tobias Harris added 15 points.

Despite their efforts, the Magic couldn't overcome the Hawks' strong performance, resulting in their 41st loss of the season. The Hawks, on the other hand, continued their impressive run, securing their 46th win.

## Rotowire prompt (Direct)

**System prompt:**
Your task is to write a summary of a basketball game whose statistics are given in JSON.
You must remain faithful to the provided statistics in your summary.
Instead of responding directly with the text, you must generate Jinja-style expressions in the text with references to fields in the provided JSON, e.g., given JSON that says:

```json
{
    "data": {
        "place_of_birth": "Ohio"
    }
}
```

then instead of generating "... was born in Ohio" you should generate "... was born in {{ data.place_of_birth }}". Note that you can only reference fields that exist in the JSON. The user will be displayed a version of this message that has the references filled in, and you should ALWAYS try to use such references instead of copying the text directly when possible.

---

**Few shot examples (optional):**

👤 **User:**
JSON:

```json
{
    "data": {
        "home_line": {
            "name": "Magic",
            "city": "Orlando",
            "wins": 6,
            "losses": 9,
            "pts": 92,
            "pts_qtr1": 18,
            "pts_qtr2": 28,
            "pts_qtr3": 21,
            "pts_qtr4": 25,
            "fg_pct": 45,
            "fg3_pct": 19,
            "ft_pct": 70,
            "ast": 18,
            "reb": 44,
            "tov": 14
        },
        "visitor_line": {
            "name": "Heat",
            "city": "Miami",
            "wins": 7,
```

```
                    "losses": 6,
                    "pts": 99,
                    "pts_qtr1": 22,
                    "pts_qtr2": 27,
                    "pts_qtr3": 19,
                    "pts_qtr4": 31,
                    "fg_pct": 52,
                    "fg3_pct": 46,
                    "ft_pct": 72,
                    "ast": 17,
                    "reb": 33,
                    "tov": 16
                },
                "box_score": {
                    "luol_deng": {
                        "player_name": "Luol Deng",
                        "first_name": "Luol",
                        "second_name": "Deng",
                        "city": "Miami",
                        "start_position": "F",
                        "min": 33,
                        "pts": 9,
                        "fgm": 4,
                        "fga": 9,
                        "fg_pct": 44,
                        "fg3m": 1,
                        "fg3a": 2,
                        "fg3_pct": 50,
                        "ftm": 0,
                        "fta": 0,
                        "ft_pct": 0,
                        "reb": 7,
                        "oreb": 1,
                        "dreb": 6,
                        "ast": 0,
                        "stl": 1,
                        "blk": 0,
                        "to": 2,
                        "pf": 3
                    },
                    ...
                }
            }
        }
    }
```

Write a summary of the game described in the JSON above. Remember to ALWAYS use Jinja-style expressions to reference the fields in the JSON whenever possible.

---

🖥️ **Assistant:**

The {{ data.visitor_line.city }} {{ data.visitor_line.name }} ({{ data.visitor_line.wins }}-{{ data.visitor_line.losses }}) took down the {{ data.home_line.city }} {{ data.home_line.name }} ({{ data.home_line.wins }}-{{ data.home_line.losses }}) {{ data.visitor_line.pts }}-{{ data.home_line.pts }} on Saturday night at the Amway Center in {{ data.home_line.city }}. The Heat were just too much for the Magic when all was said and done, as they made a nice run in the fourth to get a seven-point win. What makes the win even more impressive is that Miami did it without guard Dwyane Wade (hamstring). The Heat got plenty of production from some key players to help get the job done though. They were led in scoring by {{ data.box_score.chris_bosh.player_name }}, who put up {{ data.box_score.chris_bosh.pts }} points ({{ data.box_score.chris_bosh.fgm }}-{{ data.box_score.chris_bosh.fga }} FG,

{{ data.box_score.chris_bosh.fg3m }}-{{ data.box_score.chris_bosh.fg3a }} 3Pt, {{ data.box_score.chris_bosh.ftm }}-{{ data.box_score.chris_bosh.fta }} FT) to go with {{ data.box_score.chris_bosh.reb }} rebounds.
{{ data.box_score.mario_chalmers.player_name }} had a huge night also, scoring {{ data.box_score.mario_chalmers.pts }} points ({{ data.box_score.mario_chalmers.fgm }}-{{ data.box_score.mario_chalmers.fga }} FG, {{ data.box_score.mario_chalmers.fg3m }}-{{ data.box_score.mario_chalmers.fg3a }} 3Pt, {{ data.box_score.mario_chalmers.ftm }}-{{ data.box_score.mario_chalmers.fta }} FT) to go with {{ data.box_score.mario_chalmers.ast }} assists and {{ data.box_score.mario_chalmers.reb }} rebounds. Miami fans have to be happy to see rookie {{ data.box_score.shabazz_napier.player_name }} stepping up big, as he played extended minutes ({{ data.box_score.shabazz_napier.min }}), and scored {{ data.box_score.shabazz_napier.pts }} points ({{ data.box_score.shabazz_napier.fgm }}-{{ data.box_score.shabazz_napier.fga }} FG, {{ data.box_score.shabazz_napier.fg3m }}-{{ data.box_score.shabazz_napier.fg3a }} 3Pt, {{ data.box_score.shabazz_napier.ftm }}-{{ data.box_score.shabazz_napier.fta }} FT) to go with {{ data.box_score.shabazz_napier.reb }} rebounds, {{ data.box_score.shabazz_napier.stl }} steals and {{ data.box_score.shabazz_napier.ast }} assist. Miami has a quick turnaround, as they take on the Charlotte Hornets at home Sunday night. The Magic just couldn't get enough production to outscore the Heat on this night, as only three players hit double digits, with {{ data.box_score.nikola_vucevic.player_name }} having a monster night, scoring {{ data.box_score.nikola_vucevic.pts }} points ({{ data.box_score.nikola_vucevic.fgm }}-{{ data.box_score.nikola_vucevic.fga }} FG, {{ data.box_score.nikola_vucevic.ftm }}-{{ data.box_score.nikola_vucevic.fta }} FT), with {{ data.box_score.nikola_vucevic.reb }} rebounds. Vucevic was just dominate, and grabbed {{ data.box_score.nikola_vucevic.oreb }} offensive boards on the night. The other production came from {{ data.box_score.victor_oladipo.player_name }}, who had {{ data.box_score.victor_oladipo.pts }} points ({{ data.box_score.victor_oladipo.fgm }}-{{ data.box_score.victor_oladipo.fga }} FG, {{ data.box_score.victor_oladipo.fg3m }}-{{ data.box_score.victor_oladipo.fg3a }} 3Pt, {{ data.box_score.victor_oladipo.ftm }}-{{ data.box_score.victor_oladipo.fta }} FT), with {{ data.box_score.victor_oladipo.reb }} rebounds, {{ data.box_score.victor_oladipo.ast }} assist and {{ data.box_score.victor_oladipo.stl }} steals, as well as {{ data.box_score.evan_fournier.player_name }}, who scored {{ data.box_score.evan_fournier.pts }} points ({{ data.box_score.evan_fournier.fgm }}-{{ data.box_score.evan_fournier.fga }} FG, {{ data.box_score.evan_fournier.fg3m }}-{{ data.box_score.evan_fournier.fg3a }} 3PT) with {{ data.box_score.evan_fournier.ast }} assists and {{ data.box_score.evan_fournier.reb }} rebounds. The Magic get a night off now, before taking on the Cleveland Cavaliers on Monday.

🧑 **User:**
*Data for game to be described*

EXAMPLE INPUT
JSON:

```json
{
    "data": {
        "home_line": {
            "name": "Hawks",
            "city": "Atlanta",
            "wins": 46,
            "losses": 12,
            "pts": 95,
            "pts_qtr1": 28,
            "pts_qtr2": 14,
            "pts_qtr3": 34,
            "pts_qtr4": 19,
            "fg_pct": 44,
            "fg3_pct": 45,
            "ft_pct": 75,
            "ast": 27,
            "reb": 42,
            "tov": 18
        },
        "visitor_line": {
            "name": "Magic",
            "city": "Orlando",
            "wins": 19,
            "losses": 41,
            "pts": 88,
            "pts_qtr1": 16,
            "pts_qtr2": 29,
            "pts_qtr3": 22,
            "pts_qtr4": 21,
            "fg_pct": 45,
            "fg3_pct": 35,
            "ft_pct": 54,
            "ast": 22,
            "reb": 40,
            "tov": 17
        },
        "box_score": {
            "tobias_harris": {
                "player_name": "Tobias Harris",
                "first_name": "Tobias",
                "second_name": "Harris",
                "city": "Orlando",
                "start_position": "F",
                "min": 37,
                "pts": 15,
                "fgm": 5,
                "fga": 9,
                "fg_pct": 56,
                "fg3m": 2,
                "fg3a": 4,
                "fg3_pct": 50,
                "ftm": 3,
                "fta": 4,
                "ft_pct": 75,
                "reb": 4,
                "oreb": 0,
                "dreb": 4,
                "ast": 1,
                "stl": 2,
                "blk": 1,
                "to": 2,
                "pf": 0
            },
            ...
        }
    }
}
```

Write a summary of the game described in the JSON above. Remember to ALWAYS use Jinja-style expressions to reference the fields in the JSON whenever possible.

---

**Assistant:**
*LLM generates game summary using symbolic references.*

EXAMPLE OUTPUT (GPT-4-32K)

The `{{ data.home_line.city }}` `{{ data.home_line.name }}` (`{{ data.home_line.wins }}`-`{{ data.home_line.losses }}`) defeated the `{{ data.visitor_line.city }}` `{{ data.visitor_line.name }}` (`{{ data.visitor_line.wins }}`-`{{ data.visitor_line.losses }}`) `{{ data.home_line.pts }}`-`{{ data.visitor_line.pts }}`. The Hawks had a strong third quarter, scoring `{{ data.home_line.pts_qtr3 }}` points, which helped them secure the victory. They shot `{{ data.home_line.fg_pct }}`% from the field, `{{ data.home_line.fg3_pct }}`% from beyond the arc, and `{{ data.home_line.ft_pct }}`% from the free-throw line. They also had `{{ data.home_line.ast }}` assists, `{{ data.home_line.reb }}` rebounds, and `{{ data.home_line.tov }}` turnovers.

The Magic, on the other hand, struggled in the first quarter, scoring only `{{ data.visitor_line.pts_qtr1 }}` points. They shot `{{ data.visitor_line.fg_pct }}`% from the field, `{{ data.visitor_line.fg3_pct }}`% from three-point range, and a disappointing `{{ data.visitor_line.ft_pct }}`% from the free-throw line. They had `{{ data.visitor_line.ast }}` assists, `{{ data.visitor_line.reb }}` rebounds, and `{{ data.visitor_line.tov }}` turnovers.

For the Hawks, `{{ data.box_score.paul_millsap.player_name }}` led the team with `{{ data.box_score.paul_millsap.pts }}` points, `{{ data.box_score.paul_millsap.reb }}` rebounds, and `{{ data.box_score.paul_millsap.ast }}` assists. `{{ data.box_score.al_horford.player_name }}` also had a strong game with `{{ data.box_score.al_horford.pts }}` points and `{{ data.box_score.al_horford.reb }}` rebounds.

For the Magic, `{{ data.box_score.nikola_vucevic.player_name }}` had a standout performance with `{{ data.box_score.nikola_vucevic.pts }}` points and `{{ data.box_score.nikola_vucevic.reb }}` rebounds. `{{ data.box_score.tobias_harris.player_name }}` also contributed with `{{ data.box_score.tobias_harris.pts }}` points and `{{ data.box_score.tobias_harris.reb }}` rebounds.

## Rotowire prompt (Indirect)

**System prompt:**
Your task is to write a summary of a basketball game whose statistics are given in JSON. You must remain faithful to the provided statistics in your summary. After you generate the summary, you will be asked to rewrite it using Jinja-style expressions.

**Few shot examples (optional):**

👤 **User:**
JSON:

```json
{
    "data": {
        "home_line": {
            "name": "Magic",
            "city": "Orlando",
            "wins": 6,
            "losses": 9,
            "pts": 92,
            "pts_qtr1": 18,
            "pts_qtr2": 28,
            "pts_qtr3": 21,
            "pts_qtr4": 25,
            "fg_pct": 45,
            "fg3_pct": 19,
            "ft_pct": 70,
            "ast": 18,
            "reb": 44,
            "tov": 14
        },
        "visitor_line": {
            "name": "Heat",
            "city": "Miami",
            "wins": 7,
            "losses": 6,
            "pts": 99,
            "pts_qtr1": 22,
            "pts_qtr2": 27,
            "pts_qtr3": 19,
            "pts_qtr4": 31,
            "fg_pct": 52,
            "fg3_pct": 46,
```

```
                        "ft_pct": 72,
                        "ast": 17,
                        "reb": 33,
                        "tov": 16
                    },
                    "box_score": {
                        "luol_deng": {
                            "player_name": "Luol Deng",
                            "first_name": "Luol",
                            "second_name": "Deng",
                            "city": "Miami",
                            "start_position": "F",
                            "min": 33,
                            "pts": 9,
                            "fgm": 4,
                            "fga": 9,
                            "fg_pct": 44,
                            "fg3m": 1,
                            "fg3a": 2,
                            "fg3_pct": 50,
                            "ftm": 0,
                            "fta": 0,
                            "ft_pct": 0,
                            "reb": 7,
                            "oreb": 1,
                            "dreb": 6,
                            "ast": 0,
                            "stl": 1,
                            "blk": 0,
                            "to": 2,
                            "pf": 3
                        },
                        ...
                    }
                }
            }
```

Write a summary of the game described above.

---

🖥 **Assistant:**
The Miami Heat (7-6) took down the Orlando Magic (6-9) 99-92 on Saturday night at the Amway Center in Orlando. The Heat were just too much for the Magic when all was said and done, as they made a nice run in the fourth to get a seven-point win. What makes the win even more impressive is that Miami did it without guard Dwyane Wade (hamstring). The Heat got plenty of production from some key players to help get the job done though. They were led in scoring by Chris Bosh, who put up 32 points (13-20 FG, 4-6 3Pt, 2-2 FT) to go with 10 rebounds. Mario Chalmers had a huge night also, scoring 24 points (10-16 FG, 2-5 3Pt, 2-4 FT) to go with eight assists and five rebounds. Miami fans have to be happy to see rookie Shabazz Napier stepping up big, as he played extended minutes (28), and scored 12 points (4-8 FG, 2-5 3Pt, 2-4 FT) to go with two rebounds, two steals and one assist. Miami has a quick turnaround, as they take on the Charlotte Hornets at home Sunday night. The Magic just couldn't get enough production to outscore the Heat on this night, as only three players hit double digits, with Nikola Vucevic having a monster night, scoring 33 points (12-24 FG, 9-11 FT), with 17 rebounds. Vucevic was just dominate, and grabbed 11 offensive boards on the night. The other production came from Victor Oladipo, who had 14 points (5-11 FG, 1-4 3Pt, 3-5 FT), with four rebounds, two assist and two steals, as well as Evan Fournier, who scored 11 points (5-15 FG, 1-5 3PT) with six assists and six rebounds. The Magic get a night off now, before taking on the Cleveland Cavaliers on Monday.

👤 **User:**
Your job is to now rewrite the answer you provided above, but instead of responding directly with the text, you must generate Jinja-style expressions in the text with references to fields in the provided JSON, e.g., given JSON that says:

```json
{
    "data": {
        "place_of_birth": "Ohio"
    }
}
```

then instead of generating "... was born in Ohio" you should generate "... was born in {{ data.place_of_birth }}". Note that you can only reference fields that exist in the JSON. The user will be displayed a version of this message that has the references filled in, and you should ALWAYS try to use such references instead of copying the text directly when possible. Remember, you MUST make sure you ONLY use fields that exist in the JSON. You must NOT make up fields.

---

🖥 **Assistant:**
The {{ data.visitor_line.city }} {{ data.visitor_line.name }}
({{ data.visitor_line.wins }}-{{ data.visitor_line.losses }}) took down the
{{ data.home_line.city }} {{ data.home_line.name }}
({{ data.home_line.wins }}-{{ data.home_line.losses }})
{{ data.visitor_line.pts }}-{{ data.home_line.pts }} on Saturday night at the
Amway Center in {{ data.home_line.city }}. The Heat were just too much for the
Magic when all was said and done, as they made a nice run in the fourth to get a
seven-point win. What makes the win even more impressive is that Miami did it
without guard Dwyane Wade (hamstring). The Heat got plenty of production from
some key players to help get the job done though. They were led in scoring by
{{ data.box_score.chris_bosh.player_name }}, who put up
{{ data.box_score.chris_bosh.pts }} points
({{ data.box_score.chris_bosh.fgm }}-{{ data.box_score.chris_bosh.fga }} FG,
{{ data.box_score.chris_bosh.fg3m }}-{{ data.box_score.chris_bosh.fg3a }}
3Pt, {{ data.box_score.chris_bosh.ftm }}-{{ data.box_score.chris_bosh.fta }}
FT) to go with {{ data.box_score.chris_bosh.reb }} rebounds.
{{ data.box_score.mario_chalmers.player_name }} had a huge night also, scoring
{{ data.box_score.mario_chalmers.pts }} points ({{ data.box_score.
mario_chalmers.fgm }}-{{ data.box_score.mario_chalmers.fga }} FG,
{{ data.box_score.mario_chalmers.fg3m }}-{{ data.box_score.mario_chalmers.
fg3a }} 3Pt,
{{ data.box_score.mario_chalmers.ftm }}-{{ data.box_score.mario_chalmers.
fta }} FT) to go with {{ data.box_score.mario_chalmers.ast }} assists and
{{ data.box_score.mario_chalmers.reb }} rebounds. Miami fans have to be happy
to see rookie {{ data.box_score.shabazz_napier.player_name }} stepping up big,
as he played extended minutes ({{ data.box_score.shabazz_napier.min }}), and
scored {{ data.box_score.shabazz_napier.pts }} points ({{ data.box_score.
shabazz_napier.fgm }}-{{ data.box_score.shabazz_napier.fga }} FG,
{{ data.box_score.shabazz_napier.fg3m }}-{{ data.box_score.shabazz_napier.
fg3a }} 3Pt,
{{ data.box_score.shabazz_napier.ftm }}-{{ data.box_score.shabazz_napier.
fta }} FT) to go with {{ data.box_score.shabazz_napier.reb }} rebounds,
{{ data.box_score.shabazz_napier.stl }} steals and
{{ data.box_score.shabazz_napier.ast }} assist. Miami has a quick turnaround, as
they take on the Charlotte Hornets at home Sunday night. The Magic just couldn't
get enough production to outscore the Heat on this night, as only three players hit
double digits, with {{ data.box_score.nikola_vucevic.player_name }} having a
monster night, scoring {{ data.box_score.nikola_vucevic.pts }} points ({{ data.
box_score.nikola_vucevic.fgm }}-{{ data.box_score.nikola_vucevic.fga }} FG,
{{ data.box_score.nikola_vucevic.ftm }}-{{ data.box_score.nikola_vucevic.
fta }} FT), with {{ data.box_score.nikola_vucevic.reb }} rebounds. Vucevic was
just dominate, and grabbed {{ data.box_score.nikola_vucevic.oreb }} offensive

boards on the night. The other production came from
`{{ data.box_score.victor_oladipo.player_name }}`, who had
`{{ data.box_score.victor_oladipo.pts }}` points (`{{ data.box_score.victor_oladipo.fgm }}`-`{{ data.box_score.victor_oladipo.fga }}` FG,
`{{ data.box_score.victor_oladipo.fg3m }}`-`{{ data.box_score.victor_oladipo.fg3a }}` 3Pt,
`{{ data.box_score.victor_oladipo.ftm }}`-`{{ data.box_score.victor_oladipo.fta }}` FT), with `{{ data.box_score.victor_oladipo.reb }}` rebounds,
`{{ data.box_score.victor_oladipo.ast }}` assist and
`{{ data.box_score.victor_oladipo.stl }}` steals, as well as
`{{ data.box_score.evan_fournier.player_name }}`, who scored
`{{ data.box_score.evan_fournier.pts }}` points (`{{ data.box_score.evan_fournier.fgm }}`-`{{ data.box_score.evan_fournier.fga }}` FG,
`{{ data.box_score.evan_fournier.fg3m }}`-`{{ data.box_score.evan_fournier.fg3a }}` 3PT) with `{{ data.box_score.evan_fournier.ast }}` assists and
`{{ data.box_score.evan_fournier.reb }}` rebounds. The Magic get a night off now,
before taking on the Cleveland Cavaliers on Monday.

---

**User:**
*Data for game to be described*

EXAMPLE INPUT
JSON:

```json
{
    "data": {
        "home_line": {
            "name": "Hawks",
            "city": "Atlanta",
            "wins": 46,
            "losses": 12,
            "pts": 95,
            "pts_qtr1": 28,
            "pts_qtr2": 14,
            "pts_qtr3": 34,
            "pts_qtr4": 19,
            "fg_pct": 44,
            "fg3_pct": 45,
            "ft_pct": 75,
            "ast": 27,
            "reb": 42,
            "tov": 18
        },
        "visitor_line": {
            "name": "Magic",
            "city": "Orlando",
            "wins": 19,
            "losses": 41,
            "pts": 88,
            "pts_qtr1": 16,
            "pts_qtr2": 29,
            "pts_qtr3": 22,
            "pts_qtr4": 21,
            "fg_pct": 45,
            "fg3_pct": 35,
            "ft_pct": 54,
            "ast": 22,
            "reb": 40,
            "tov": 17
        },
        "box_score": {
            "tobias_harris": {
                "player_name": "Tobias Harris",
                "first_name": "Tobias",
                "second_name": "Harris",
                "city": "Orlando",
                "start_position": "F",
                "min": 37,
                "pts": 15,
                "fgm": 5,
                "fga": 9,
                "fg_pct": 56,
                "fg3m": 2,
                "fg3a": 4,
                "fg3_pct": 50,
                "ftm": 3,
                "fta": 4,
                "ft_pct": 75,
                "reb": 4,
                "oreb": 0,
                "dreb": 4,
                "ast": 1,
                "stl": 2,
                "blk": 1,
                "to": 2,
                "pf": 0
            },
            ...
        }
    }
}
```

Write a summary of the game described above.

🖥️ **Assistant:**
*LLM generates summary without symbolic references.*

EXAMPLE OUTPUT (GPT-4-32K)
The Atlanta Hawks (46-12) defeated the Orlando Magic (19-41) 95-88 at home. The Hawks had a strong performance in the third quarter, scoring 34 points, which helped them secure the victory. The Hawks were led by Paul Millsap, who scored 20 points, and Al Horford, who added 17 points and grabbed 13 rebounds. Jeff Teague also contributed with 17 points and 7 assists. On the other hand, the Magic were led by Nikola Vucevic, who had a strong performance with 21 points and 15 rebounds. Victor Oladipo also contributed with 19 points, while Tobias Harris added 15 points. Despite the loss, the Magic outperformed the Hawks in the second quarter, scoring 29 points.

---

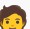 **User:**
Your job is to now rewrite the answer you provided above, but instead of responding directly with the text, you must generate Jinja-style expressions in the text with references to fields in the provided JSON, e.g., given JSON that says:

```json
{
    "data": {
        "place_of_birth": "Ohio"
    }
}
```

then instead of generating "... was born in Ohio" you should generate "... was born in {{ data.place_of_birth }}". Note that you can only reference fields that exist in the JSON. The user will be displayed a version of this message that has the references filled in, and you should ALWAYS try to use such references instead of copying the text directly when possible. Remember, you MUST make sure you ONLY use fields that exist in the JSON. You must NOT make up fields.

🖥 **Assistant:**
*LLM rewrites summary to contain symbolic references.*

EXAMPLE OUTPUT (GPT-4-32K)
The {{ data.home_line.city }} {{ data.home_line.name }} ({{ data.home_line.wins }}-{{ data.home_line.losses }}) defeated the {{ data.visitor_line.city }} {{ data.visitor_line.name }} ({{ data.visitor_line.wins }}-{{ data.visitor_line.losses }}) {{ data.home_line.pts }}-{{ data.visitor_line.pts }} at home. The Hawks had a strong performance in the third quarter, scoring {{ data.home_line.pts_qtr3 }} points, which helped them secure the victory. The Hawks were led by {{ data.box_score.paul_millsap.player_name }}, who scored {{ data.box_score.paul_millsap.pts }} points, and {{ data.box_score.al_horford.player_name }}, who added {{ data.box_score.al_horford.pts }} points and grabbed {{ data.box_score.al_horford.reb }} rebounds. {{ data.box_score.jeff_teague.player_name }} also contributed with {{ data.box_score.jeff_teague.pts }} points and {{ data.box_score.jeff_teague.ast }} assists. On the other hand, the Magic were led by {{ data.box_score.nikola_vucevic.player_name }}, who had a strong performance with {{ data.box_score.nikola_vucevic.pts }} points and {{ data.box_score.nikola_vucevic.reb }} rebounds. {{ data.box_score.victor_oladipo.player_name }} also contributed with {{ data.box_score.victor_oladipo.pts }} points, while {{ data.box_score.tobias_harris.player_name }} added {{ data.box_score.tobias_harris.pts }} points. Despite the loss, the Magic outperformed the Hawks in the second quarter, scoring {{ data.visitor_line.pts_qtr2 }} points.

### I.2.3 Obituary prompts

**Obituary prompt (Baseline)**

**System prompt:**
Your task is to write an obituary about a person whose description is given in JSON. You must mention and be faithful about every property of the person in your obituary.

---

**Few shot examples (optional):**

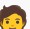 **User:**
JSON:

```json
{
    "data": {
        "name": "Niels Henrik Abel",
```

```json
            "age": "27,
            "place_of_birth": "Nedstrand",
            "place_of_death": "Froland",
            "father": "Sren Georg Abel",
            "country_of_citizenship": "Norway",
            "educated_at": "University of Oslo",
            "field_of_work": "calculus",
            "occupation": "mathematician",
            "employer": "University of Oslo",
            "religion_or_worldview": "Lutheranism",
            "award_received": "Grand prix des sciences
                mathematiques",
            "member_of": "Royal Norwegian Society of Sciences and
                Letters",
            "cause_of_death": "tuberculosis",
            "residence": "Norway",
            "notable_work": "abelian group",
            "languages_spoken": "Norwegian"
        }
    }
    ```
```

---

**🖥 Assistant:**

Niels Henrik Abel, a renowned mathematician, passed away at the young age of 27 in Froland, Norway. Born in Nedstrand, Abel was a proud Norwegian citizen and a devoted Lutheran. He was educated at the University of Oslo, where he later worked as a professor. Abel made significant contributions to the field of calculus, most notably the concept of the abelian group. He was a member of the Royal Norwegian Society of Sciences and Letters and received the prestigious Grand prix des sciences mathematiques. Sadly, Abel's life was cut short due to tuberculosis. He will be remembered for his remarkable achievements and dedication to the world of mathematics.

---

*One more few-shot example like the one above (hidden to save space)*

---

**🧑 User:**

*Data for obituary to be generated*

EXAMPLE INPUT
JSON:

```json
{
    "data": {
        "name": "Herbert Spencer",
        "age": 83,
        "place_of_birth": "Derby",
        "place_of_death": "Brighton",
        "father": "William George Spencer",
        "country_of_citizenship": "United Kingdom of Great Britain and Ireland",
        "field_of_work": "botany",
        "occupation": "writer",
        "employer": "The Economist",
        "place_of_burial": "Highgate Cemetery",
        "religion_or_worldview": "agnosticism",
        "member_of": "Athenaeum Club",
        "notable_work": "Social Statics",
        "nominated_for": "Nobel Peace Prize",
        "languages_spoken": "English"
    }
}
```

---

**🖥 Assistant:**

*Generate untemplated obituary. LLM generates obituary without symbolic references*

EXAMPLE OUTPUT (GPT-3.5-4K)

Herbert Spencer, a distinguished writer and botanist, has passed away at the age of 83 in Brighton. Born in Derby to William George Spencer, he was a citizen of the United Kingdom of Great Britain and Ireland. Spencer made significant contributions to the field of botany and worked as a writer for The Economist. He was a member of the Athenaeum Club and was known for his notable work, "Social Statics." Spencer was a proponent of agnosticism and was fluent in English. Although not awarded, he was nominated for the Nobel Peace Prize. Spencer will be laid to rest at Highgate Cemetery, leaving behind a legacy of intellectual pursuits and contributions to the scientific community.

## Obituary prompt (Direct)

**System prompt:**

Your task is to write an obituary about a person whose description is given in JSON. You must mention and be faithful about every property of the person in your obituary.

Instead of responding directly with the text, you must generate Jinja-style expressions in the text with references to fields in the provided JSON, e.g., given JSON that says:

```json
{
    "data": {
        "place_of_birth": "Ohio"
    }
}
```

then instead of generating "... was born in Ohio" you should generate "... was born in `{{ data.place_of_birth }}`". Note that you can only reference fields that exist in the JSON. The user will be displayed a version of this message that has the references filled in, and you should ALWAYS try to use such references instead of copying the text directly when possible.

---

**Few shot examples (optional):**

👦 **User:**

JSON:

```json
{
    "data": {
        "name": "Niels Henrik Abel",
        "age": 27,
        "place_of_birth": "Nedstrand",
        "place_of_death": "Froland",
        "father": "Sren Georg Abel",
        "country_of_citizenship": "Norway",
        "educated_at": "University of Oslo",
        "field_of_work": "calculus",
        "occupation": "mathematician",
        "employer": "University of Oslo",
        "religion_or_worldview": "Lutheranism",
        "award_received": "Grand prix des sciences
            mathematiques",
        "member_of": "Royal Norwegian Society of Sciences and
            Letters",
        "cause_of_death": "tuberculosis",
        "residence": "Norway",
        "notable_work": "abelian group",
        "languages_spoken": "Norwegian"
    }
}
```

---

🖥 **Assistant:**

{{ data.name }}, a renowned {{ data.occupation }}, passed away at the young age of {{ data.age }} in {{ data.place_of_death }}, {{ data.country_of_citizenship }}. Born in {{ data.place_of_birth }}, {{ data.name.split(' ')[2] }} was a proud Norwegian citizen and a devoted Lutheran. He was educated at the {{ data.educated_at }}, where he later worked as a professor. {{ data.name.split(' ')[2] }} made significant contributions to the field of {{ data.field_of_work }}, most notably the concept of the {{ data.notable_work }}. He was a member of the {{ data.member_of }} and received the prestigious {{ data.award_received }}. Sadly, {{ data.name.split(' ')[2] }}'s life was cut short due to {{ data.cause_of_death }}. He will be remembered for his remarkable achievements and dedication to the world of mathematics.

*One more few-shot example like the one above (hidden to save space)*

🧑 **User:**
*Data for obituary to be generated*

```
EXAMPLE INPUT
JSON:

```json
{
    "data": {
        "name": "Herbert Spencer",
        "age": 83,
        "place_of_birth": "Derby",
        "place_of_death": "Brighton",
        "father": "William George Spencer",
        "country_of_citizenship": "United Kingdom of Great Britain and Ireland",
        "field_of_work": "botany",
        "occupation": "writer",
        "employer": "The Economist",
        "place_of_burial": "Highgate Cemetery",
        "religion_or_worldview": "agnosticism",
        "member_of": "Athenaeum Club",
        "notable_work": "Social Statics",
        "nominated_for": "Nobel Peace Prize",
        "languages_spoken": "English"
    }
}
```
```

🖥️ **Assistant:**
*LLM directly generates obituary using symbolic references.*

```
EXAMPLE OUTPUT (GPT-3.5-4K)
{{ data.name }}, a prominent writer and botanist, has passed away at the age of {{ data.age }} in {{ data.place_of_death }}. Born in {{ data.place_of_birth }} to father {{ data.father }}, {{ data.name.split(' ')[1] }} was a citizen of the {{ data.country_of_citizenship }}. He dedicated his life to the field of {{ data.field_of_work }} and worked as a writer for {{ data.employer }}. {{ data.name.split(' ')[1] }} was a member of the {{ data.member_of }} and was nominated for the {{ data.nominated_for }}. He was fluent in {{ data.languages_spoken }} and was known for his notable work, "{{ data.notable_work }}." {{ data.name.split(' ')[1] }}'s agnostic worldview shaped his perspective on life and the world. He will be laid to rest at {{ data.place_of_burial }}. His contributions to the field of botany and his literary works will be remembered for generations to come.
```

## I.3    Obituary QA evaluation

**Prompt for obituary QA evaluation**

**System prompt:**
Your task is to answer questions about an obituary. Do not rely on any knowledge other than what is provided in the obituary. You should state your answer without any explanation. The answer can be a single word or a short phrase. The user message will be of the form:

STORY:
<story>
QUESTION: <question>
You should reply only with your answer. If you do not know the answer, you should reply with 'Unknown.'

---

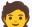 **User:**
*Question to be answered, alongside story*

> EXAMPLE INPUT
> STORY:
> Herbert Spencer, a distinguished pharmacist and author of "Social Statics," has passed away at the age of 83 in Vienna. Born in Derby, Spencer was the son of Lord Byron and a citizen of the United Kingdom of Great Britain and Ireland. Although his field of work was primarily in physics, he made significant contributions as a pharmacist and worked for The Economist. Spencer was a member of the Athenaeum Club and was known for his agnostic worldview. He was nominated for the Nobel Prize in Physiology or Medicine for his notable work. Spencer will be laid to rest at Highgate Cemetery. He was fluent in Swedish and will be remembered for his remarkable achievements in the field of pharmacy and his influential writings.
> QUESTION: Where was Herbert Spencer born?

---

🖥️ **Assistant:**
*LLM generates answer to question based on provided obituary.*

> EXAMPLE OUTPUT (HAND-WRITTEN FOR ILLUSTRATIVE PURPOSES)
> Derby

## I.3.1 Financial prompts

---

**Financial prompt (Baseline)**

**System prompt:**
Your task is to answer a finance-related question based on some company information given in JSON. You must remain remain faithful to the provided information in your answer.

---

**Few shot examples (optional):**

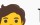 **User:**
QUESTION: How do the 50-d moving averages between AMZN and TSM compare?
JSON:

```json
{
    "data": {
        "AMZN": {
            "Symbol": "AMZN",
            "AssetType": "Common Stock",
            "Name": "Amazon.com Inc",
            "Description": "Amazon.com, Inc. is an American
                multinational technology company which focuses
                on e-commerce, cloud computing, digital
                streaming, and artificial intelligence. It is
                one of the Big Five companies in the U.S.
                information technology industry, along with
                Google, Apple, Microsoft, and Facebook. The
                company has been referred to as one of the most
                 influential economic and cultural forces in
                the world, as well as the world's most valuable
                brand.",
            "CIK": 1018724,
            "Exchange": "NASDAQ",
            "Currency": "USD",
            "Country": "USA",
```

```
                "Sector": "TRADE & SERVICES",
                "Industry": "RETAIL-CATALOG & MAIL-ORDER HOUSES",
                "Address": "410 TERRY AVENUE NORTH, SEATTLE, WA,
                    US",
                "FiscalYearEnd": "December",
                "LatestQuarter": "2023-09-30",
                "MarketCapitalization": 1235158172000,
                "EBITDA": 63431000000,
                "PERatio": 94.9,
                "PEGRatio": 11.85,
                "BookValue": 16.35,
                "DividendPerShare": 0.0,
                "DividendYield": 0.0,
                "EPS": 1.26,
                "RevenuePerShareTTM": 52.56,
                "ProfitMargin": 0.0243,
                "OperatingMarginTTM": 0.0572,
                "ReturnOnAssetsTTM": 0.0247,
                "ReturnOnEquityTTM": 0.0872,
                "RevenueTTM": 538046005000,
                "GrossProfitTTM": 225152000000,
                "DilutedEPSTTM": 1.26,
                "QuarterlyEarningsGrowthYOY": -0.986,
                "QuarterlyRevenueGrowthYOY": 0.108,
                "AnalystTargetPrice": 160.61,
                "TrailingPE": 94.9,
                "ForwardPE": 43.86,
                "PriceToSalesRatioTTM": 1.797,
                "PriceToBookRatio": 6.61,
                "EVToRevenue": 1.948,
                "EVToEBITDA": 19.3,
                "Beta": 1.188,
                "52WeekHigh": 145.86,
                "52WeekLow": 81.43,
                "50DayMovingAverage": 132.31,
                "200DayMovingAverage": 117.12,
                "SharesOutstanding": 10317800000,
                "DividendDate": "None",
                "ExDividendDate": "None"
            },
            "TSM": {
                "Symbol": "TSM",
                "AssetType": "Common Stock",
                ...
                "DividendDate": "2024-01-11",
                "ExDividendDate": "2023-12-14"
            }
        }
    }
    ```
```

*Two more few-shot example like the one above (hidden to save space)*

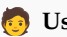 **User:**
*Question and data required to answer it.*

EXAMPLE INPUT
QUESTION: Which company has the largest market capitalization, GOOG or ASML?
JSON:

```json
{
    "data": {
        "ASML": {
            "Symbol": "ASML",
            "AssetType": "Common Stock",
            "Name": "ASML Holding NV ADR",
            "Description": "ASML Holding NV develops, produces, markets, sells and services
                    advanced semiconductor equipment systems consisting of lithography,
                    metrology and inspection related systems for memory and logic chip
                    manufacturers. The company is headquartered in Veldhoven, the Netherlands.",
            "CIK": 937966,
            "Exchange": "NASDAQ",
            "Currency": "USD",
            "Country": "USA",
            "Sector": "TECHNOLOGY",
            "Industry": "SPECIAL INDUSTRY MACHINERY, NEC",
            "Address": "DE RUN 6501, DR VELDHOVEN, NL",
            "FiscalYearEnd": "December",
            "LatestQuarter": "2023-09-30",
            "MarketCapitalization": 240264151000,
            "EBITDA": 9398800000,
            "PERatio": 28.83,
            "PEGRatio": 1.703,
            "BookValue": 30.49,
            "DividendPerShare": 4.43,
            "DividendYield": 0.0111,
            "EPS": 20.6,
            "RevenuePerShareTTM": 1.224,
            "ProfitMargin": 0.284,
            "OperatingMarginTTM": 0.327,
            "ReturnOnAssetsTTM": 0.156,
            "ReturnOnEquityTTM": 0.763,
            "RevenueTTM": 26751701000,
            "GrossProfitTTM": 10700100000,
            "DilutedEPSTTM": 20.6,
            "QuarterlyEarningsGrowthYOY": 0.121,
            "QuarterlyRevenueGrowthYOY": 0.155,
            "AnalystTargetPrice": 680.53,
            "TrailingPE": 28.83,
            "ForwardPE": 30.49,
            "PriceToSalesRatioTTM": 12.43,
            "PriceToBookRatio": 29.61,
            "EVToRevenue": 12.62,
            "EVToEBITDA": 35.86,
            "Beta": 1.168,
            "52WeekHigh": 770.52,
            "52WeekLow": 434.41,
            "50DayMovingAverage": 615.52,
            "200DayMovingAverage": 657.88,
            "SharesOutstanding": 393400000,
            "DividendDate": "2023-11-10",
            "ExDividendDate": "2023-11-01"
        },
        "GOOG": {
            "Symbol": "GOOG",
            "AssetType": "Common Stock",
            ...
            "DividendDate": "None",
            "ExDividendDate": "None"
        }
    }
}
```

🖥️ **Assistant:**
*LLM generates answer without symbolic references.*

EXAMPLE OUTPUT (GPT-4-32K)
The company with the largest market capitalization is Alphabet Inc Class C (GOOG) with a market capitalization of
1,536,753,926,000 USD.

---

**Financial prompt (Direct)**

**System prompt:**
Your task is to answer a finance-related question based on some company information given in JSON. You must remain remain faithful to the provided information in your answer. Instead of responding directly with the text, you must generate Jinja-style expressions in the text with references to fields in the provided JSON, e.g., given JSON that says:

```json
{
    "data": {
        "place_of_birth": "Ohio"
    }
}
```

then instead of generating "... was born in Ohio" you should generate "... was born in {{ data.place_of_birth }}". Note that you can only reference fields that exist in the JSON. The user will be displayed a version of this message that has the references filled in, and you should ALWAYS try to use such references instead of copying the text directly when possible.

---

**Few shot examples (optional):**

👤 **User:**
QUESTION: How do the 50-d moving averages between AMZN and TSM compare?
JSON:

```json
{
    "data": {
        "AMZN": {
            "Symbol": "AMZN",
            "AssetType": "Common Stock",
            "Name": "Amazon.com Inc",
            "Description": "Amazon.com, Inc. is an American
                multinational technology company which focuses
                on e-commerce, cloud computing, digital
                streaming, and artificial intelligence. It is
                one of the Big Five companies in the U.S.
                information technology industry, along with
                Google, Apple, Microsoft, and Facebook. The
                company has been referred to as one of the most
                 influential economic and cultural forces in
                the world, as well as the world's most valuable
                 brand.",
            "CIK": 1018724,
            "Exchange": "NASDAQ",
            "Currency": "USD",
            "Country": "USA",
            "Sector": "TRADE & SERVICES",
            "Industry": "RETAIL-CATALOG & MAIL-ORDER HOUSES",
            "Address": "410 TERRY AVENUE NORTH, SEATTLE, WA,
                US",
            "FiscalYearEnd": "December",
            "LatestQuarter": "2023-09-30",
            "MarketCapitalization": 1235158172000,
            "EBITDA": 63431000000,
            "PERatio": 94.9,
            "PEGRatio": 11.85,
            "BookValue": 16.35,
            "DividendPerShare": 0.0,
            "DividendYield": 0.0,
            "EPS": 1.26,
```

```
                "RevenuePerShareTTM": 52.56,
                "ProfitMargin": 0.0243,
                "OperatingMarginTTM": 0.0572,
                "ReturnOnAssetsTTM": 0.0247,
                "ReturnOnEquityTTM": 0.0872,
                "RevenueTTM": 538046005000,
                "GrossProfitTTM": 225152000000,
                "DilutedEPSTTM": 1.26,
                "QuarterlyEarningsGrowthYOY": -0.986,
                "QuarterlyRevenueGrowthYOY": 0.108,
                "AnalystTargetPrice": 160.61,
                "TrailingPE": 94.9,
                "ForwardPE": 43.86,
                "PriceToSalesRatioTTM": 1.797,
                "PriceToBookRatio": 6.61,
                "EVToRevenue": 1.948,
                "EVToEBITDA": 19.3,
                "Beta": 1.188,
                "52WeekHigh": 145.86,
                "52WeekLow": 81.43,
                "50DayMovingAverage": 132.31,
                "200DayMovingAverage": 117.12,
                "SharesOutstanding": 10317800000,
                "DividendDate": "None",
                "ExDividendDate": "None"
            },
            "TSM": {
                "Symbol": "TSM",
                "AssetType": "Common Stock",
                ...
                "DividendDate": "2024-01-11",
                "ExDividendDate": "2023-12-14"
            }
        }
    }
    ```
```

---

🖥 **Assistant:**
The 50-day moving average for {{ data.AMZN.Name }} ({{ data.AMZN.Symbol }}) is
{{ data.AMZN['50DayMovingAverage'] }} USD, while for {{ data.TSM.Name }}
({{ data.TSM.Symbol }}) it is {{ data.TSM['50DayMovingAverage'] }} USD.
Therefore, {{ data.AMZN.Symbol }}'s 50-day moving average is higher than
{{ data.TSM.Symbol }}'s.

---

*Two more few-shot example like the one above (hidden to save space)*

---

🧑 **User:**
*Question and data required to answer it.*

EXAMPLE INPUT
QUESTION: Which company has the largest market capitalization, GOOG or ASML?
JSON:

```json
{
    "data": {
        "ASML": {
            "Symbol": "ASML",
            "AssetType": "Common Stock",
            "Name": "ASML Holding NV ADR",
            "Description": "ASML Holding NV develops, produces, markets, sells and services
                    advanced semiconductor equipment systems consisting of lithography,
                    metrology and inspection related systems for memory and logic chip
                    manufacturers. The company is headquartered in Veldhoven, the Netherlands.",
            "CIK": 937966,
            "Exchange": "NASDAQ",
            "Currency": "USD",
            "Country": "USA",
            "Sector": "TECHNOLOGY",
            "Industry": "SPECIAL INDUSTRY MACHINERY, NEC",
            "Address": "DE RUN 6501, DR VELDHOVEN, NL",
            "FiscalYearEnd": "December",
            "LatestQuarter": "2023-09-30",
            "MarketCapitalization": 240264151000,
            "EBITDA": 9398800000,
            "PERatio": 28.83,
            "PEGRatio": 1.703,
            "BookValue": 30.49,
            "DividendPerShare": 4.43,
            "DividendYield": 0.0111,
            "EPS": 20.6,
            "RevenuePerShareTTM": 1.224,
            "ProfitMargin": 0.284,
            "OperatingMarginTTM": 0.327,
            "ReturnOnAssetsTTM": 0.156,
            "ReturnOnEquityTTM": 0.763,
            "RevenueTTM": 26751701000,
            "GrossProfitTTM": 10700100000,
            "DilutedEPSTTM": 20.6,
            "QuarterlyEarningsGrowthYOY": 0.121,
            "QuarterlyRevenueGrowthYOY": 0.155,
            "AnalystTargetPrice": 680.53,
            "TrailingPE": 28.83,
            "ForwardPE": 30.49,
            "PriceToSalesRatioTTM": 12.43,
            "PriceToBookRatio": 29.61,
            "EVToRevenue": 12.62,
            "EVToEBITDA": 35.86,
            "Beta": 1.168,
            "52WeekHigh": 770.52,
            "52WeekLow": 434.41,
            "50DayMovingAverage": 615.52,
            "200DayMovingAverage": 657.88,
            "SharesOutstanding": 393400000,
            "DividendDate": "2023-11-10",
            "ExDividendDate": "2023-11-01"
        },
        "GOOG": {
            "Symbol": "GOOG",
            "AssetType": "Common Stock",
            ...
            "DividendDate": "None",
            "ExDividendDate": "None"
        }
    }
}
```

🖥 **Assistant:**
*LLM generates answer directly with symbolic references.*

EXAMPLE OUTPUT (GPT-4-32K):
The company with the largest market capitalization is `{{ data.GOOG.Name }}` (`{{ data.GOOG.Symbol }}`) with a market capitalization of `{{ data.GOOG.MarketCapitalization }}` USD.

**Financial prompt (Indirect)**

**System prompt:**
Your task is to answer a finance-related question based on some company information given in JSON. You must remain remain faithful to the provided information in your answer.
After you generate the answer, you will be asked to rewrite it using Jinja-style expressions.

---

**Few shot examples (optional):**

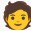 **User:**
QUESTION: How do the 50-d moving averages between AMZN and TSM compare?
JSON:

```json
{
    "data": {
        "AMZN": {
            "Symbol": "AMZN",
            "AssetType": "Common Stock",
            "Name": "Amazon.com Inc",
            "Description": "Amazon.com, Inc. is an American
                multinational technology company which focuses
                on e-commerce, cloud computing, digital
                streaming, and artificial intelligence. It is
                one of the Big Five companies in the U.S.
                information technology industry, along with
                Google, Apple, Microsoft, and Facebook. The
                company has been referred to as one of the most
                 influential economic and cultural forces in
                the world, as well as the world's most valuable
                 brand.",
            "CIK": 1018724,
            "Exchange": "NASDAQ",
            "Currency": "USD",
            "Country": "USA",
            "Sector": "TRADE & SERVICES",
            "Industry": "RETAIL-CATALOG & MAIL-ORDER HOUSES",
            "Address": "410 TERRY AVENUE NORTH, SEATTLE, WA,
                US",
            "FiscalYearEnd": "December",
            "LatestQuarter": "2023-09-30",
            "MarketCapitalization": 1235158172000,
            "EBITDA": 63431000000,
            "PERatio": 94.9,
            "PEGRatio": 11.85,
            "BookValue": 16.35,
            "DividendPerShare": 0.0,
            "DividendYield": 0.0,
            "EPS": 1.26,
            "RevenuePerShareTTM": 52.56,
            "ProfitMargin": 0.0243,
            "OperatingMarginTTM": 0.0572,
            "ReturnOnAssetsTTM": 0.0247,
            "ReturnOnEquityTTM": 0.0872,
            "RevenueTTM": 538046005000,
            "GrossProfitTTM": 225152000000,
            "DilutedEPSTTM": 1.26,
            "QuarterlyEarningsGrowthYOY": -0.986,
            "QuarterlyRevenueGrowthYOY": 0.108,
            "AnalystTargetPrice": 160.61,
            "TrailingPE": 94.9,
            "ForwardPE": 43.86,
```

```
                "PriceToSalesRatioTTM": 1.797,
                "PriceToBookRatio": 6.61,
                "EVToRevenue": 1.948,
                "EVToEBITDA": 19.3,
                "Beta": 1.188,
                "52WeekHigh": 145.86,
                "52WeekLow": 81.43,
                "50DayMovingAverage": 132.31,
                "200DayMovingAverage": 117.12,
                "SharesOutstanding": 10317800000,
                "DividendDate": "None",
                "ExDividendDate": "None"
            },
            "TSM": {
                "Symbol": "TSM",
                "AssetType": "Common Stock",
                ...
                "DividendDate": "2024-01-11",
                "ExDividendDate": "2023-12-14"
            }
        }
    }
    ```
```

🖥 **Assistant:**
The 50-day moving average for Amazon.com Inc (AMZN) is 132.31 USD, while for
Taiwan Semiconductor Manufacturing (TSM) it is 90.20 USD. Therefore, AMZN's
50-day moving average is higher than TSM's.

🧑 **User:**
Your job is to now rewrite the answer you provided above, but instead of responding
directly with the text, you must generate Jinja-style expressions in the text with
references to fields in the provided JSON, e.g., given JSON that says:

```json
{
    "data": {
        "place_of_birth": "Ohio"
    }
}
```

then instead of generating "... was born in Ohio" you should generate "... was born in
`{{ data.place_of_birth }}`". Note that you can only reference fields that exist in the
JSON. The user will be displayed a version of this message that has the references
filled in, and you should ALWAYS try to use such references instead of copying the
text directly when possible. Remember, you MUST make sure you ONLY use fields
that exist in the JSON. You must NOT make up fields.

🖥 **Assistant:**
The 50-day moving average for `{{ data.AMZN.Name }}` (`{{ data.AMZN.Symbol }}`) is
`{{ data.AMZN['50DayMovingAverage'] }}` USD, while for `{{ data.TSM.Name }}`
(`{{ data.TSM.Symbol }}`) it is `{{ data.TSM['50DayMovingAverage'] }}` USD.
Therefore, `{{ data.AMZN.Symbol }}`'s 50-day moving average is higher than
`{{ data.TSM.Symbol }}`'s.

*Two more few-shot example like the one above (hidden to save space)*

🧑 **User:**
*Question and data required to answer it*

EXAMPLE INPUT
QUESTION: Which company has the largest market capitalization, GOOG or ASML?
JSON:

```json
{
    "data": {
        "ASML": {
            "Symbol": "ASML",
            "AssetType": "Common Stock",
            "Name": "ASML Holding NV ADR",
            "Description": "ASML Holding NV develops, produces, markets, sells and services
                    advanced semiconductor equipment systems consisting of lithography,
                    metrology and inspection related systems for memory and logic chip
                    manufacturers. The company is headquartered in Veldhoven, the Netherlands.",
            "CIK": 937966,
            "Exchange": "NASDAQ",
            "Currency": "USD",
            "Country": "USA",
            "Sector": "TECHNOLOGY",
            "Industry": "SPECIAL INDUSTRY MACHINERY, NEC",
            "Address": "DE RUN 6501, DR VELDHOVEN, NL",
            "FiscalYearEnd": "December",
            "LatestQuarter": "2023-09-30",
            "MarketCapitalization": 240264151000,
            "EBITDA": 9398800000,
            "PERatio": 28.83,
            "PEGRatio": 1.703,
            "BookValue": 30.49,
            "DividendPerShare": 4.43,
            "DividendYield": 0.0111,
            "EPS": 20.6,
            "RevenuePerShareTTM": 1.224,
            "ProfitMargin": 0.284,
            "OperatingMarginTTM": 0.327,
            "ReturnOnAssetsTTM": 0.156,
            "ReturnOnEquityTTM": 0.763,
            "RevenueTTM": 26751701000,
            "GrossProfitTTM": 10700100000,
            "DilutedEPSTTM": 20.6,
            "QuarterlyEarningsGrowthYOY": 0.121,
            "QuarterlyRevenueGrowthYOY": 0.155,
            "AnalystTargetPrice": 680.53,
            "TrailingPE": 28.83,
            "ForwardPE": 30.49,
            "PriceToSalesRatioTTM": 12.43,
            "PriceToBookRatio": 29.61,
            "EVToRevenue": 12.62,
            "EVToEBITDA": 35.86,
            "Beta": 1.168,
            "52WeekHigh": 770.52,
            "52WeekLow": 434.41,
            "50DayMovingAverage": 615.52,
            "200DayMovingAverage": 657.88,
            "SharesOutstanding": 393400000,
            "DividendDate": "2023-11-10",
            "ExDividendDate": "2023-11-01"
        },
        "GOOG": {
            "Symbol": "GOOG",
            "AssetType": "Common Stock",
            ...
            "DividendDate": "None",
            "ExDividendDate": "None"
        }
    }
}
```

🖥 **Assistant:**
*LLM generates answer without symbolic references.*

EXAMPLE OUTPUT (GPT-4-32K)
The company with the largest market capitalization is Alphabet Inc Class C (GOOG) with a market capitalization of 1,536,753,926,000 USD.

👤 **User:**
Your job is to now rewrite the answer you provided above, but instead of responding directly with the text, you must generate Jinja-style expressions in the text with references to fields in the provided JSON, e.g., given JSON that says:

```json
{
    "data": {
        "place_of_birth": "Ohio"
    }
}
```

then instead of generating "... was born in Ohio" you should generate "... was born in {{ data.place_of_birth }}". Note that you can only reference fields that exist in the JSON. The user will be displayed a version of this message that has the references filled in, and you should ALWAYS try to use such references instead of copying the text directly when possible. Remember, you MUST make sure you ONLY use fields that exist in the JSON. You must NOT make up fields.

---

💻 **Assistant:**
*LLM rewrites answer using symbolic references.*

> EXAMPLE OUTPUT (GPT-4-32K)
> The company with the largest market capitalization is {{ data.GOOG.Name }} ({{ data.GOOG.Symbol }}) with a market capitalization of {{ data.GOOG.MarketCapitalization }} USD.

## I.3.2 GSM8K prompts

### GSM8K prompt (Direct)

**System prompt:**
You are a helpful assistant.

---

**Few shot examples (optional):**

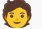 **User:**
We are working on solving a math question. We want to generate the computation steps in jinja templating format along the way. We allow using `{{` and `}}` to quote variables, and `{% set variable=value %}` to set variables. We do NOT allow using setting unknown variables (e.g., X) to a value, and the computation should be as simple as possible.
Let's do a practice round.

---

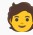 **Assistant:**
Sounds great!

---

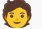 **User:**
Question: There are 15 trees in the grove. Grove workers will plant trees in the grove today. After they are done, there will be 21 trees. How many trees did the grove workers plant today?

---

💻 **Assistant:**
There are 15 trees originally{% set initial_trees = 15 %}. Then there were 21 trees after some more were planted{% set final_trees = 21 %}. Therefore, there are {% set trees_planted = final_trees - initial_trees %}{{ trees_planted }} trees planted today.
Answer: {{ trees_planted }}

---

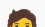 **User:**
Question: If there are 3 cars in the parking lot and 2 more cars arrive, how many cars are in the parking lot?

🖥️ **Assistant:**
There are originally 3 cars{% set initial_cars = 3 %}. 2 more cars
arrive{% set arrived_cars = 2 %}. The total car in the parking lot is
{% set final_cars = initial_cars + arrived_cars %}{{ final_cars }}.
Answer: {{ final_cars }}

---

🧑 **User:**
Question: Leah had 32 chocolates and her sister had 42. If they ate 35, how many
pieces do they have left in total?

---

🖥️ **Assistant:**
Originally, Leah had 32 chocolates{% set leah_chocolates = 32 %}, and her sister
had 42{% set sister_chocolates = 42 %}. In total, they had
{% set total_chocolates = leah_chocolates + sister_chocolates %}{{
total_chocolates }} chocolates. After eating 35{% set chocolates_eaten = 35 %},
they had {% set chocolates_left = total_chocolates - chocolates_eaten %}{{
chocolates_left }} left.
Answer: {{ chocolates_left }}

---

🧑 **User:**
Question: Jason had 20 lollipops. He gave Denny some lollipops. Now Jason has 12
lollipops. How many lollipops did Jason give to Denny?

---

🖥️ **Assistant:**
Jason started with 20 lollipops{% set initial_lollipops = 20 %}. Then he had 12
after giving some to Denny. Now Jason has 12
lollipops{% set final_lollipops = 12 %}. So he gave Denny
{% set lollipops_given = initial_lollipops - final_lollipops %}{{
lollipops_given }}.
Answer: {{ lollipops_given }}

---

🧑 **User:**
Question: Shawn has five toys. For Christmas, he got two toys each from his mom
and dad. How many toys does he have now?

---

🖥️ **Assistant:**
Shawn started with 5 toys{% set initial_toys = 5 %}. He got 2 toys each from his
mom and dad, that's {% set new_toys = 2 + 2 %}{{ new_toys }} more toys. So he
have {% set toys_now = initial_toys + new_toys %}{{ toys_now }} toys.
Answer: {{ toys_now }}

---

🧑 **User:**
Question: There were nine computers in the server room. Five more computers were
installed each day, from monday to thursday. How many computers are now in the
server room?

---

🖥️ **Assistant:**
There were originally 9 computers{% set initial_computers = 9 %}. For monday
to thursday{% set days = 4 %}, 5 more computers were
added{% set daily_added_computers = 5 %}. Therefore, a total of
{% set total_added_computers = daily_added_computers * days %}{{
total_added_computers }} is added. There are
{% set total_computers = total_added_computers + initial_computers %}{{
total_computers }} in the server room?
Answer: {{ total_computers }}

---

🧑 **User:**

Question: Michael had 58 golf balls. On tuesday, he lost 23 golf balls. On wednesday, he lost 2 more. How many golf balls did he have at the end of wednesday?

💻 **Assistant:**
Michael started with 58 golf balls`{% set initial_golf_balls = 58 %}`. After losing 23 on tuesday`{% set lost_tuesday = 23 %}`, he had `{% set golf_balls_left = initial_golf_balls - lost_tuesday %}``{{ golf_balls_left }}`. He lost 2 more on wednesday`{% set lost_wednesday = 2 %}`, so he has `{% set final_golf_balls = golf_balls_left - lost_wednesday %}``{{ final_golf_balls }}` golf balls.
Answer: `{{ final_golf_balls }}`

🧑 **User:**
Question: Olivia has $23. She bought five bagels for $3 each. How much money does she have left?

💻 **Assistant:**
Olivia had 23 dollars`{% set initial_money = 23 %}`. 5 bagels`{% set bagel_count = 5 %}` for 3 dollars each`{% set bagel_price = 3 %}` will be `{% set bagel_cost = bagel_price * bagel_count %}``{{ total_bagel_cost }}` dollars. So she has `{% set money_left = initial_money - bagel_cost %}``{{ money_left }}` dollars.
Answer: `{{ money_left }}`

🧑 **User:**
*Question we want an answer for*

> EXAMPLE INPUT
> Here is another one:
> Question: Janet's ducks lay 16 eggs per day. She eats three for breakfast every morning and bakes muffins for her friends every day with four. She sells the remainder at the farmers' market daily for $2 per fresh duck egg. How much in dollars does she make every day at the farmers' market?

💻 **Assistant:**
*LLM generates answer with symbolic references*

> EXAMPLE OUTPUT (GPT-4-8K)
> Janet's ducks lay 16 eggs per day`{% set total_eggs = 16 %}`. She eats 3 for breakfast`{% set breakfast_eggs = 3 %}` and uses 4 for muffins`{% set muffin_eggs = 4 %}`. So she uses
> `{% set used_eggs = breakfast_eggs + muffin_eggs %}``{{ used_eggs }}` eggs. The remainder is
> `{% set market_eggs = total_eggs - used_eggs %}``{{ market_eggs }}` eggs. She sells these at the farmers' market for $2 per egg`{% set egg_price = 2 %}`, so she makes `{% set market_income = market_eggs * egg_price %}``{{ market_income }}` dollars every day.
> Answer: `{{ market_income }}`

