# OpenReview forum: "Towards Verifiable Text Generation with Symbolic References"
_colmweb.org/COLM/2024/Conference — COLM_

### Official Review · Reviewer_oNbF · 2024-05-11

**Rating:** 6
**Confidence:** 4
**Ethics Flag:** 1

**Summary:**

The authors propose a new prompting technique to generate textual descriptions
grounded in structured data, e.g., a news blurb describing financial data or
sports statistics. The key innovation appears to be the use of Jinja-like syntax
to inline symbol references in the generated text. The authors evaluate this
approach on several data sources: biographies from WikiData, sports data from
Rotowire and financial data from a financial API, where they observe that their
proposed approach generates descriptions that are comparably fluent and grounded
to a baseline approach. The key benefit however are the references which can be
used to show readers links to the source data: through a user study, the authors
demonstrate that these links allow readers to verify the data presented in the
text more quickly and easily.

Finally, the authors show two other ways in which the symbolic references could be useful:
first to help language models override their priors with counterfactual data,
and second as a way to help language models use symbolic expressions for math problems.
Unfortunately, in both of these applications, the results are mixed and are not
consistently better than the baselines.

**Questions To Authors:**

- In the discussion of the counterfactual evaluation, you mention that some of
  the difference in performance might be explained as a precision-recall trade
  off. This seems easily quantifiable, have you measured precision/recall of the
  generated answers?
- It is great that you included dataset sizes in the appendix, but I think this
  information belongs in the main paper. It is really hard to understand what's
  a meaningful improvement without that information. Given the relatively small
  sizes of the evaluation datasets, could you do a statistical analysis and
  report which results are actually statistically significant?

**Reasons To Accept:**

- The paper is well-written, and is comprehensive, with quantitative and human
  evaluations, some qualitative error analysis and detailed information on the
  prompts used.
- The main contribution of this paper appears to be showing that their prompt
  strategy works in providing references to structured data. The authors
  convincingly show that this makes it easier for users to verify the generated
  text through a user study, though they do not show that it significantly
  improves the groundedness of the generations.

**Reasons To Reject:**

There are several dimensions along which this paper makes contributions, but
unfortunately, I found it hard to find one in which it excels:

- Application: The chief contribution of this paper appears to be generating
  symbolic references along with text to help show those references to users.
  While valuable, the application has been proposed several times in the
  literature, and this paper limits its scope to referencing (near-)exact
  matches from the source data. I think this too narrow, given the state of
  language models today (e.g., the GPT3.5/4 models used in this paper).

- Technique: While the paper provides a clever way to prompt language
  models to generate text with grounded symbolic references, this technique isn't
  particularly effective at improving the groundedness of the generations over
  baselines (which are already very good). Furthermore, given the relatively
  small evaluation set sizes, it is unclear if any gains are statistically
  significant or not.

- Data: the paper does generate its own synthetic counterfactual obituaries
  dataset, but the authors do not really analyze this dataset or make it public.

---

> ### Author Rebuttal · Authors · 2024-05-31
>
> We would like to thank reviewers for their effort and constructive feedback! We are encouraged that XpW4 and oNbF acknowledge our contribution and find our experiments comprehensive and convincing.
>
> Given this paper is in the **LMs and interactions track**, we aim to tackle **the pressing problem of human verification of LLM generations**. We developed **SymGen to make it easier for users’ to verify an LLM’s generation** and evaluate its performance via multiple human studies. To thoroughly understand how SymGen works on different tasks, we also experiment to ensure **SymGen’s generation quality is on par with the baseline** on multiple tasks, and we **do not intend to claim that SymGen can improve generations** (e.g., better reasoning or grounded generation).
>
> ---
>
> We will release the obituary dataset.
>
> > Not improving groundedness
>
> We don't intend to improve the groundedness of the generation but focus on whether SymGen can maintain the generation quality compared to the baselines.
>
> > referencing near-exact matches
>
> We focus on end users verification of the generation; studying the references to near-exact matches is a first step to this problem and avoids other confounding factors like incorrect user judgment on similar answers.
>
> > the application has been proposed
>
> While data-to-text generation is well-established, to the best of our knowledge we are the first to focus on how the generation can be made easier to verify by users.
>
> > precision/recall of generated answers in counterfactual eval
>
> We mentioned that SymGen tends to not generate the response more frequently if it doesn’t know the answer, and conjectured that SymGen may in a loose sense optimize for higher precision than recall of the answer. One possible explanation for this is that symbolic generation includes slightly less information than regular generation (esp. in the few-shot case with higher unknown rate), which leads to more incorrect answers (more questions were unanswered). The results in Table 6 can be construed as the precision of SymGen, but computing recall is non-trivial in our setup.
>
> > report statistic significance
>
> As mentioned above, we focus on the human eval dataset in the rebuttal. We recruited 60 annotators for the human study, which is a reasonably large size for human evaluation of this kind. We also add standard error on the mean for table 7 below; note that our time, confidence, and effort results have non-overlapping error bars. https://imgur.com/a/5IWhJbn

---

> > ### Comment · Reviewer_oNbF · 2024-06-06
> > **Thank you for your response**
> >
> > Thanks for your detailed response and for clarifying that the scope of the contributions. My concerns mostly related to whether the methods proposed here could improve the factuality of responses, which might be less relevant for the LM and interactions track. I find the prompting method to be quite novel and the user study convincing. I have increased my score to marginally above the acceptance threshold.

---

### Official Review · Reviewer_oZFv · 2024-05-11

**Rating:** 4
**Confidence:** 4
**Ethics Flag:** 1

**Summary:**

The paper presents a novel approach, symbolically grounded generation (SymGen), which enhances the capabilities of large language models (LLMs) for complex reasoning tasks. The authors demonstrate that SymGen can produce text with accurate symbolic references while maintaining fluency and factuality, which is a significant advancement in the field of natural language processing.

Please note that I doubt whether the font used in this paper aligns with the Call for Papers (CFP) guidelines for COLM.

**Questions To Authors:**

Please see Reasons to Reject

**Reasons To Accept:**

1. Innovative Approach: The paper introduces a new method for text generation that significantly improves verifiability without compromising the quality of the text generated by LLMs.

2. Empirical Evidence: Strong empirical results support the claims made by the authors, with improvements demonstrated across a range of data-to-text and question-answering experiments.

3. Practical Application: The proposed SymGen method has practical implications for applications such as virtual assistants and knowledge bases, where the ability to generate verifiable text is crucial.

4. Human Study: The inclusion of a human study adds weight to the findings, showing that the annotations provided by SymGen can streamline the verification process.

5. Well-Structured Paper: The paper adheres to the COLM reviewing guidelines and is presented in a clear, accessible manner, making it a valuable contribution to the literature.

**Reasons To Reject:**

1. Limited Dataset Scope and Insignificant Superiority: The evaluation is based on a small number of toy datasets, which may not fully capture the diversity of real-world applications. I expect to see the performance of the method on some more realistic and complex datasets, e.g., table reasoning, where the method may be superior.

2. Complexity of Indirect SymGen: The indirect SymGen approach, while maintaining text quality, could be more costly and less reliable with certain LLMs, which might be a concern for practical adoption.

3. Error Analysis: The paper could benefit from a more detailed error analysis, particularly in understanding the types and causes of errors in symbolic generation.

4. Generalizability: It is unclear how well the proposed method would generalize to other types of structured data (e.g., Graph, Table, Databases, etc.) and language models, which is a key consideration for its broader acceptance.

5. More than Verifiability: The authors emphasize the proposed method's verifiability. But what are the benefits? I am curious about future work. Does better verifiability make it easier for the LLMs to receive human feedback and evolve? Do the authors have more in-depth insights?

6. No Training: The paper does not attempt to train LLMs to master the technique of generating symbolic references, which limits its impact.

---

### Official Review · Reviewer_XpW4 · 2024-05-11

**Rating:** 6
**Confidence:** 5
**Ethics Flag:** 1

**Summary:**

The authors propose SymGen which provides symbolic references to the input structure (JSON) along with the generation. Two variants are proposed, direct and indirect.  The direct approach prompts LLM to generate text along with Jinja-like references in one go. The in-direct approach first prompts LLM to generate a text summary, that is subsequently transformed into text with symbolic references by prompting.
The experiments:
1. study changes in textual quality due to symbolic reference generation
2. show that the annotations cannot be simply replaced by regex-based replacement.
3. show the system improves the verifiability speed.
4. show that SymGen can also be applied to QA on structured (JSON) data and reasoning setup.

**Questions To Authors:**

In section 3.1 : Why does it say "incomplete JSONS"?

**Reasons To Accept:**

I believe that such a system is a natural attempt towards this problem. That could mean that the learnings are benificial to the community. I also appreciate that the authors have conducted multiple experiments that answer a variety of questions.

**Reasons To Reject:**

In general, there is no clear recommendation by the authors. I see many mixed results, which is fine, but the paper does not map it to clear insights or error analysis.

Table 1 (a) and (b) have mixed results depending on the dataset. The paper claims:
1. That rotowire is a difficult dataset and requires a lot of exact numbers replicated. However, the baseline results do not support this.
2. SynthBio has a simpler JSON compared to Rotowire, but the authors suggest that biographical generation in symbolic space is slightly harder. It would be very helpful to have some error analysis to gain an understanding of inherent difficulty.

The authors claim that the indirect strategy does not work for GPT3.5 as it is weaker than GPT-4. Due to a lack of examples/comments, it is difficult to say why is that the case. It could be helpful if there were different and possibly open-source models used as well to be able to helpfully distill when this method works. Although I do acknowledge the GPU requirements issue here.

There are no details on how the parsing and representation would work for representation other than JSON. Hence, it seems that the proposed method is limited to JSONs. Furthermore, there is a limited description of the parsing stage itself apart from seemingly simple regex + replacement.

It is not obvious from the writeup why the authors chose not to use the in-direct method for the experiments in Table 2.

For the QA experiments, the gap between GPT-3.5 baseline vs direct methods is huge compared to GPT4, and experiments in Table 1. My assumption is that it could be due to shorter outputs. But the error rate is missing in this experiment which makes the results unclear.

In the related work section authors talk about LLM-based verification methods, without justifying the differences or why comparisons cannot be made. On a related note, and agreeing with the associated post-processing cost of prompting, regex-based baseline is not a fair comparison. A prompt-based baseline would be more adequate. An approach could be (only to cite possibilities): for each summary sentence, prompt which key-value pairs support the sentence? Or a prompt: where all sentences are passed as a list at once to make it comparable to indirect generation.

---

> ### Author Rebuttal · Authors · 2024-05-31
>
> > Rotowire is a difficult dataset and requires a lot of exact numbers replicated [but] the baseline results do not support this.
>
> Rotowire is a challenging testbed due to the sheer size of the JSON data presented to the LLM (see App. H.2.2).
>
> > SynthBio has a simpler JSON compared to Rotowire, but the authors suggest [it] is slightly harder. It would be very helpful to have some error analysis [...]
>
> For SynthBio, JSON values are not perfectly formatted, meaning that LLMs will either output grammatically incorrect text or have to perform post-processing on-the-fly via more complex templates, which increases the chance of errors. We will add some examples to the paper.
>
> > the indirect strategy does not work for GPT3.5 as it is weaker than GPT-4 [but why?]
>
> We found that for many examples GPT-3.5 would ignore the instructions/in-context examples and directly output a response with symbolic references in the first step. Though the metrics were still reasonable (e.g., for SynthBio, BLEU was 32.67 for 0-shot and 33.10 for 2-shot), we omitted these results from the table since GPT-3.5 was not correctly executing the indirect strategy.
>
> > no details on how the parsing and representation would work [...] other than JSON. [...] limited description of the parsing stage
>
> In short, we use an off-the-shelf Jinja parser. Further, we found our approach to work fairly well by simply adjusting the prompt as long as the data format and templating language are not too exotic (e.g., YAML and Python f-strings also work).
>
> > not obvious [...]  why the authors chose not to use the in-direct method [in] Table 2.
>
> Since the QA evaluation was particularly costly, we chose to focus on a single LLM for this dataset–GPT-3.5–for which the indirect strategy does not function properly.
>
> > For the QA experiments, the gap between GPT-3.5 baseline vs direct methods is huge compared to GPT4, and experiments in Table 1.
>
> We found that GPT-3.5 struggled to do CoT-like behavior when operating in symbolic space. This is discussed in more detail in App. F (financial error analysis).
>
> > [the] regex-based baseline is not a fair comparison. A prompt-based baseline would be more adequate
>
> Our goal with the Regex baseline was to ascertain whether the benefits of SymGen generations can be obtained without using LLMs. There may be other ways of augmenting generations with information that may be useful for verifying their correctness, but we believe such a comparison is out-of-scope for this paper.

---

> > ### Comment · Reviewer_XpW4 · 2024-06-04
> > **Thank you for the response**
> >
> > Thank you for the response and clarifications. Given the author's reply, it appears that not all concerns about the write-up can be resolved without altering the text significantly; therefore, I will be keeping the scores unchanged.

---

> > > ### Author Response · Authors · 2024-06-07
> > > **Thank you for your comment and clarifications**
> > >
> > > Thank you for your response!
> > >
> > > Firstly in case you haven’t seen the overall review response message (we weren't able to include the following in the response to your review due to character limit, but included it in the response for reviewer oNbF):
> > >
> > > ---
> > > We would like to thank reviewers for their effort and constructive feedback! We are encouraged that XpW4 and oNbF acknowledge our contribution and find our experiments comprehensive and convincing.
> > >
> > > Given this paper is in the **LMs and interactions track**, we aim to tackle the pressing problem of human verification of LLM generations. We developed **SymGen to make it easier for users’ to verify an LLM’s generation** and evaluate its performance via multiple human studies. To thoroughly understand how SymGen works on different tasks, we also experiment to ensure **SymGen’s generation quality is on par with the baseline** on multiple tasks, and we do not intend to claim that SymGen can improve generations (e.g., better reasoning or grounded generation).
> > >
> > > ---
> > >
> > > Thank you for the detailed questions clarifying the results and model choices. We will implement the following changes to clarify these details in our paper:
> > > - Describe the parsing pipeline in depth and mention the opportunities for using other formats (e.g., yaml).
> > > - Clarify why we use specific strategy is adopted in experiments in table 1 and 2, and additional description and examples for the Rotowire and Synthbio dataset.
> > > - Clarify the point "In the related work section authors talk about LLM-based verification methods, without justifying the differences or why comparisons cannot be made.": While data-to-text generation is well-established, to the best of our knowledge we are the first to focus on how the generation can be made easier to verify by users.
> > >
> > > The proposed changes are mostly clarifying experimental details–which can be easily incorporated in this version without major alterations to the text. Most importantly, we don't see this as changing the main claim of our paper: that SymGen can make generations easily verifiable by humans. For this reason, we kindly ask you to reconsider increasing your score and please let us know if you find additional changes are necessary. Thanks!

---

### Author Response · Authors · 2024-06-07
**Overall comment and planned paper updates**

We would like to thank all reviewers for their time, effort, and constructive feedback! We are encouraged that both reviewer XpW4 and oNbF acknowledge the contribution of our system for providing references in generation and reducing the effort for verification, find our experiments comprehensive and results convincing, and recognize the value and benefits of our methods to the community.

Most of the reviewers' concern centered around whether SymGen can lead to better text quality: we’ve addressed the concerns in the comments for each reviewer individually. But we want to flag that our work centers on the topic of **LMs and interactions**, and our paper is scoped to tackle the pressing problem of human verification of LLM generations, and both reviewers XpW4 and oNbF appreciate the method we propose and the multiple experiments we conduct to validate the method.

We will update our paper following their suggestions. Here is the list of the planned updates based on reviewer feedback:
- Highlight our paper’s key contribution for reducing users’ effort for verifying the outputs. (XpW4, oNbF)
- Describe the parsing pipeline in depth and mention the opportunities for using other formats (e.g., yaml). (XpW4)
- Clarify why we use specific strategy is adopted in experiments in table 1 and 2, and additional description and examples for the Rotowire and Synthbio dataset. (XpW4)
- Update the discussion on “precision/recall of generated answers in counterfactual eval“ (oNbF)
- Report statistic significance for table 7  (https://imgur.com/a/5IWhJbn, oNbF)
- Update the related work section to reflect that our work is among the first to focus on how LLM generations can be made easier to verify by users in data-to-text generation settings. (XpW4, oNbF)

We find the suggested changes for the camera-ready version are not substantial (mostly clarifications), and we can incorporate the changes in the final draft within this review cycle.

---

### Decision · Program_Chairs · 2024-07-10

**Decision:**

Accept

**Comment:**

Overall:  This paper suggests an approach for data-to-text generation with symbolic references as an intermediate step that can be presented to humans to make it easier to verify the text. The approach is interesting and the paper is generally well-written with comprehensive experiments.  There was a common confusion amongst reviewers about some of the goals of this paper, especially regarding the comparison of quality in Section 3.  I think that this can mostly be fixed in revisions, but will require some significant rewriting.  I also would suggest that Section 4 should be expanded further since it seems like the main goal of this approach is to help humans with verification which is what’s investigated by the user study.

Strengths:
- Interesting prompting approach
- helping users verify information (annotators took less time, and felt more confident and easier to verify outputs)
- Adding the SymGen annotations doesn’t seem to degrade output quality by much and they show this on three datasets

Weaknesses:
- The goals and contributions of this paper seemed to have been a point of confusion for both XpW4 and oNbF, though they seemed more satisfied after reading the author responses. Authors clarified in the response that “developed SymGen to make it easier for users’ to verify an LLM’s generation and evaluate its performance via multiple human studies”.  Authors also did state this in the paper: that the “main goal is not to improve upon the baseline”.  But I think this needs to be emphasized more clearly in general. Since this is actually the main goal of the work, my suggestion to the authors is that they may want to condense section 3 and instead do more of a deep dive into section 4 with more analysis and visualizations about the human studies.  Note: this seems mostly a writing/clarity issue but may require bigger revisions.
- Some lingering questions about experimental setup, baselines, etc. (see Reviewer XpW4)  that need to be clarified within the paper.  ← Authors have offered to update many of these in their general comments at the top of the discussion section.
- Experimental results seem a bit mixed (brought up by XpW4) without clear conclusions. I think that this may be ok for the intended purpose of this paper, which as I understand it, is to show that these annotations help with verification without degrading the quality by much. But, again, the writing needs to be updated to make the goals of this exploration more clear.

[At least one review was discounted during the decision process due to quality]